# META-DYNAMICAL STATE SPACE MODELS FOR INTEGRATIVE NEURAL DATA ANALYSIS

**Ayesha Vermani**[1], **Josue Nassar**[2], **Hyungju Jeon**[1], **Matthew Dowling**[1], **Il Memming Park**[1]

[1] Champalimaud Centre for the Unknown, Champalimaud Foundation, Portugal
[2] RyvivyR, USA
`{ayesha.vermani, memming.park}@research.fchampalimaud.org`

## ABSTRACT

Learning shared structure across environments facilitates rapid learning and adaptive behavior in neural systems. This has been widely demonstrated and applied in machine learning to train models that are capable of generalizing to novel settings. However, there has been limited work exploiting the shared structure in neural activity during similar tasks for learning latent dynamics from neural recordings. Existing approaches are designed to infer dynamics from a single dataset and cannot be readily adapted to account for statistical heterogeneities across recordings. In this work, we hypothesize that similar tasks admit a corresponding family of related solutions and propose a novel approach for meta-learning this solution space from task-related neural activity of trained animals. Specifically, we capture the variabilities across recordings on a low-dimensional manifold which concisely parametrizes this family of dynamics, thereby facilitating rapid learning of latent dynamics given new recordings. We demonstrate the efficacy of our approach on few-shot reconstruction and forecasting of synthetic dynamical systems, and neural recordings from the motor cortex during different arm reaching tasks.

## 1 INTRODUCTION

Latent variable models are widely used in neuroscience to extract dynamical structure underlying high-dimensional neural activity (Pandarinath et al., 2018; Schimel et al., 2022; Dowling et al., 2024). While latent dynamics provide valuable insights into behavior and generate testable hypotheses of neural computation (Luo et al., 2023; Nair et al., 2023), they are typically inferred from a single recording session. As a result, these models are sensitive to small variations in the underlying dynamics and exhibit limited generalization capabilities. In parallel, a large body of work in machine learning has focused on training models from diverse datasets that can rapidly adapt to novel settings. However, there has been limited work on inferring generalizable dynamical systems from data, with existing approaches mainly applied to settings with known low-dimensional dynamics (Yin et al., 2021; Kirchmeyer et al., 2022).

Integrating noisy neural recordings from different animals and/or tasks for learning the underlying dynamics presents a unique set of challenges. This is partly due to heterogeneities in recordings across sessions such as the number and tuning properties of recorded neurons, as well as different stimulus statistics and behavioral modalities across cognitive tasks. This challenge is further compounded by the lack of inductive biases for disentangling the variabilities across dynamics into shared and dataset-specific components. Recent evidence suggests that learned latent dynamics underlying activity of task-trained biological and artificial neural networks demonstrate similarities when engaged in related tasks (Gallego et al., 2018; Maheswaranathan et al., 2019; Safaie et al., 2023). In a related line of work, neural networks trained to perform multiple cognitive tasks with overlapping cognitive components learn to reuse dynamical motifs, thereby facilitating few-shot adaptation on novel tasks (Turner & Barak, 2023; Driscoll et al., 2024).

Motivated by these observations, we propose a novel framework for meta-learning latent dynamics from neural recordings (Vermani et al., 2024a). Our approach is to encode the variations in the latent dynamical structure present across neural recordings in a low-dimensional vector, $e \in \mathbb{R}^{d_e}$, which we refer to as the *dynamical embedding*. During training, the model learns to adapt a common

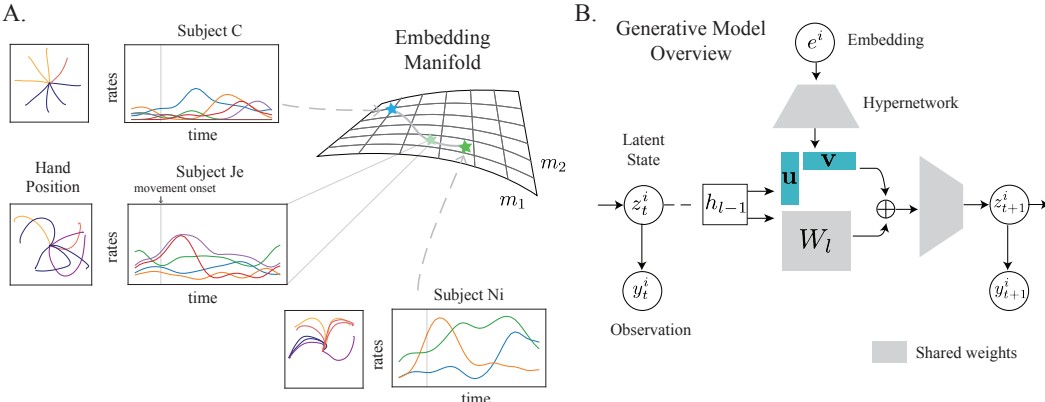

Figure 1: **A.** Neural recordings display heterogeneities in the number and tuning properties of recorded neurons and reflect diverse behavioral responses. The low-dimensional embedding manifold captures this diversity in dynamics. **B.** Our method learns to adapt a common latent dynamics conditioned on the embedding via low-rank changes to the model parameters.

latent dynamical system model conditioned on the dynamical embedding. We learn the dynamical embedding manifold from a diverse collection of neural recordings, allowing rapid learning of latent dynamics in the analysis of data-limited regime commonly encountered in neuroscience experiments.

Our contributions can be summarized as follows:

1. We propose a novel parameterization of latent dynamics that facilitates integration and learning of meta-structure over diverse neural recordings.

2. We develop an inference scheme to jointly infer the embedding and latent state trajectory, as well as the corresponding dynamics model directly from data.

3. We demonstrate the efficacy of our method on few-shot reconstruction and forecasting for synthetic datasets and motor cortex recordings obtained during different reaching tasks.

## 2   CHALLENGES WITH JOINTLY LEARNING DYNAMICS ACROSS DATASETS

Neurons from different sessions and/or subjects are partially observed, non-overlapping and exhibit diverse response properties. Even chronic recordings from a single subject exhibit drift in neural tuning over time (Driscoll et al., 2017). Moreover, non-simultaneously recorded neural activity lack pairwise correspondence between single trials. This makes joint inference of latent states and learning the corresponding latent dynamics by integrating different recordings ill-posed and highly non-trivial.

As an illustrative example, let's consider a case where these recordings exhibit oscillatory latent dynamics with variable velocities (Fig. 2A). One possible strategy for jointly inferring the dynamics from these recordings is learning a shared dynamics model, along with dataset-specific likelihood functions that map these dynamics to individual recordings (Pandarinath et al., 2018).

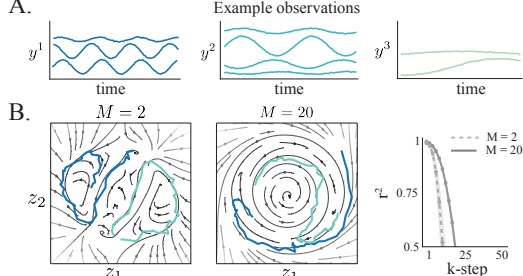

Figure 2: **A.** Three different example neural recordings, where the speed of the latent dynamics varies across them. **B.** One generative model is trained on $M = 2$ or $M = 20$ datasets. While increasing the number of datasets allows the model to learn limit cycle, it is unable to capture the different speeds leading to poor forecasting performance.

However, without additional inductive biases, this strategy does not generally perform well when there are variabilities in the underlying dynamics. Specifically, when learning dynamics from two example datasets ($M = 2$), we observed that a model

with shared dynamics either learned separate solutions or overfit to one dataset, obscuring global structure across recordings (Fig. 2A). When we increased the diversity of training data ($M = 20$), the dynamics exhibited a more coherent global structure, albeit with an overlapping solution space (Fig. 2B). As a result, this model had poor forecasting performance of neural activity in both cases, which is evident in the k-step $r^2$ (Fig. 2B). While we have a priori knowledge of the source of variations in dynamics for this example, this is typically not the case with real neural recordings. Therefore, we develop an approach for inferring the variation across recordings and use it to define a solution space of related dynamical systems (Fig. 1A).

## 3   INTEGRATING NEURAL RECORDINGS FOR META-LEARNING DYNAMICS

Let $y_{1:T}^{1:M}$ denote neural time series datasets of length $T$, with $y_t^i \in \mathbb{R}^{d_{y^i}}$, collected from $M$ different sessions and/or subjects performing related tasks. We are interested in learning a generative model that can jointly describe the evolution of the latent states across these datasets and rapidly adapt to novel datasets from limited trajectories. In this work, we focus on nonlinear state-space models (SSM), a powerful class of generative models for spatio-temporal datasets. An SSM is described via the following pair of equations (we drop the superscript for ease of presentation),

$$z_t \,|\, z_{t-1} \sim p_\theta(z_t \,|\, z_{t-1}), \tag{1}$$

$$y_t \,|\, z_t \sim p_\phi(y_t \,|\, z_t), \tag{2}$$

where $z_t \in \mathbb{R}^{d_z}$ is the latent state at time $t$, $p_\theta(z_t \,|\, z_{t-1})$ is the dynamics model and $p_\phi(y_t \,|\, z_t)$ is the likelihood function that maps the latent state to observed data.

We parametrize the dynamics as a Gaussian distribution $p_\theta(z_t \,|\, z_{t-1}) = \mathcal{N}(z_t \,|\, f_\theta(z_{t-1}), Q)$, where the mean is modeled by a deep neural network (DNN) $f_\theta$ and $Q$ is the covariance matrix[1]. As previous work has shown that highly expressive likelihood and dynamics can cause optimization issues (Bowman et al., 2015), we model the mean of the likelihood as an affine function of $z_t$. For instance, the likelihood for real-valued observations is defined as $p_\phi(y_t \,|\, z_t) = \mathcal{N}(y_t \,|\, Cz_t + D, R)$.

### 3.1   HIERARCHICAL STATE-SPACE MODEL FOR MULTIPLE DATASETS

We introduce a hierarchical structure in the latent dynamical system model to capture variations across datasets and jointly describe the spatiotemporal evolution across $M$ neural recordings in a unified SSM. A natural choice for learning this generative model is a fully Bayesian approach, where each dataset would have its own latent dynamics, parameterized by $\theta^i$, and a hierarchical prior would tie these dataset-specific parameters to shared parameters, $\theta \sim p(\theta)$ (Linderman et al., 2019), leading to the following SSM,

$$\theta^i \,|\, \theta \sim p(\theta^i \,|\, \theta), \tag{3}$$

$$z_t^i \,|\, z_{t-1}^i, \theta^i \sim \mathcal{N}(z_t^i \,|\, f_{\theta^i}(z_{t-1}^i), Q^i), \tag{4}$$

$$y_t^i \,|\, z_t^i \sim p_{\phi^i}(y_t^i \,|\, z_t^i), \tag{5}$$

where dataset specific likelihoods, $p_{\phi^i}(y_t^i \,|\, z_t^i)$, are used to account for different dimensionality and/or recording modality. If we assume $p(\theta^i \,|\, \theta)$ is Gaussian, i.e., $p(\theta^i \,|\, \theta) = \mathcal{N}(\theta^i \,|\, \theta, \Sigma)$, we can equivalently express the dynamics for the hierarchical generative model as,

$$\varepsilon^i \sim \mathcal{N}(\varepsilon^i \,|\, 0, \Sigma), \tag{6}$$

$$z_t^i \,|\, z_{t-1}^i, \theta, \varepsilon^i \sim \mathcal{N}\left(z_t^i \,|\, f_{\theta+\varepsilon^i}(z_{t-1}^i), Q^i\right), \tag{7}$$

where the dataset-specific dynamics parameter, $\theta^i$, is expressed as a sum of the shared parameters, $\theta$, and a dataset-specific term, $\varepsilon^i$. While this formulation is intuitive, the latent dynamics are approximated using a DNN, which introduces a substantial number of parameters and constrains scalability. To address these limitations, we propose a modified hierarchical framework that significantly improves both scalability and parameter efficiency, making it suitable for large-scale settings.

Specifically, we introduce a low-dimensional latent variable, $e^i \in \mathbb{R}^{d_e}, \mathbb{R}^{d_e} \ll \mathbb{R}^{d_\varepsilon}$—which we refer to as the dynamical embedding—that encodes dynamical variations across datasets (Rusu et al.,

---

[1]We note that $Q$ can also be parameterized via a neural network as well.

2019). This dataset-specific dynamical embedding subsequently maps to the parameter space of the latent dynamics function via a hypernetwork (Ha et al., 2016), $h_\vartheta : \mathbb{R}^{d_e^i} \to \mathbb{R}^{d_\varepsilon^i}$. Apart from improving scalability, this formulation also facilitates efficient few-shot learning since it requires simply inferring the embedding given trials from novel recordings. The generative model for this hierarchical SSM is then described as,

$$e^i \sim \mathcal{N}(0, I), \tag{8}$$

$$\theta^i = \theta + h_\vartheta(e^i), \tag{9}$$

$$z_t^i \,|\, z_{t-1}^i, e^i \sim \mathcal{N}(z_t^i \,|\, f_{\theta^i}(z_{t-1}^i), Q^i), \tag{10}$$

$$y_t^i \,|\, z_t^i \sim p_{\phi^i}(y_t^i \,|\, z_t^i), \tag{11}$$

where we drop the prior over the shared dynamics parameter, $\theta$, significantly reducing the dimensionality of the inference problem. Similar to the hierarchical Bayesian model, all datasets share the same latent dynamics, $\theta$, with the dataset-specific variation captured by the dynamical embedding, $e_i$.

We encourage learning of shared dynamical structure and further improve parameter efficiency by constraining $h_\vartheta$ to make low-rank changes to the parameters of $f_\theta$ (Fig. 1B). For example, if we parameterize $f_\theta$ as a 2-layer fully-connected neural network and constrain the hypernetwork to only make rank $d_r$ changes to the hidden weights, then $f_{\theta^i}$ would be expressed as,

$$f_{\theta^i}(z_t^i) = \mathbf{W}_o \; \sigma\big(\{\mathbf{W}_{hh} \; + \; \underbrace{h_\vartheta(e^i)}_{\text{embedding modification}}\} \sigma(\mathbf{W}_{\text{in}} \; z_t^i)\big) \tag{12}$$

$$= \underbrace{\mathbf{W}_o}_{\mathbb{R}^{d_z \times d_2}} \sigma\big(\{\underbrace{\mathbf{W}_{hh}}_{\mathbb{R}^{d_2 \times d_1}} + \underbrace{\mathbf{u}_\vartheta(e^i)}_{\mathbb{R}^{d_2 \times d_r}} \cdot \underbrace{\mathbf{v}_\vartheta(e^i)^\top}_{\mathbb{R}^{d_r \times d_1}}\} \sigma(\underbrace{\mathbf{W}_{\text{in}}}_{\mathbb{R}^{d_1 \times d_z}} z_t^i)\big) \tag{13}$$

where $\sigma(\cdot)$ denotes a point-nonlinearity, and the two functions $\mathbf{v}_\vartheta(e^i) : \mathbb{R}^d_e \to \mathbb{R}^{d_1 \times d_r}$, $\mathbf{u}_\vartheta(e^i) : \mathbb{R}^d_e \to \mathbb{R}^{d_2 \times d_r}$ map the embedding representation to form the low-rank perturbations. Both $\mathbf{u}_\vartheta$ and $\mathbf{v}_\vartheta$ are parametrized via a neural network.

## 3.2 Inference and Learning

Given $y_{1:T}^{1:M}$, we want to infer both the latent states, $z_{1:T}^{1:M}$ and the dynamical embeddings, $e^{1:M} = [e^1, \ldots, e^M]$ as well as learn the parameters of the generative model, $\Theta = \{\theta, \vartheta, \phi^1, \ldots, \phi^M\}$. Exact inference and learning requires computing the posterior, $p_\Theta(z_{1:T}^{1:M}, e^{1:M} \,|\, y_{1:T}^{1:M})$, and log marginal likelihood, $\log p_\Theta(y_{1:T}^{1:M})$, which are both intractable.

In this paper, we use a sequential variational autoencoder—an extension of variational autoencoders for state-space models—specifically, the Deep Kalman Filter (DKF) (Krishnan et al., 2015), to circumvent this issue. In order to learn the generative model, we maximize a lower-bound to the log marginal likelihood (commonly referred to as the ELBO). The ELBO for $y_{1:T}^{1:M}$ is defined as follows (trial indices are omitted for ease of notation),

$$\mathcal{L}(y_{1:T}^{1:M}) = \sum_{t,i} \mathbb{E}_{q_{\alpha,\beta}} \big[\log p_{\phi^i}(y_t^i \,|\, z_t^i)\big]$$
$$\quad - \mathbb{E}_{q_\beta} \big[\mathbb{D}_{\text{KL}}\big(q_\beta(z_t^i \,|\, \bar{y}_{1:T}^i, e^i) \,\big|\big|\, p_{\theta,\vartheta}(z_t^i \,|\, z_{t-1}^i, e^i)\big)\big] - \mathbb{D}_{\text{KL}}\big(q_\alpha(e^i \,|\, \bar{y}_{1:T}^i) \,\big|\big|\, p(e^i)\big) \tag{14}$$

where $q_\alpha$ and $q_\beta$ are encoders that approximate the posterior distributions over the dynamical embedding and latent state for dataset $i$, respectively, and the joint expectation factorizes as $\mathbb{E}_{q_{\alpha,\beta}} \equiv \mathbb{E}_{q_\beta(z_t^i|\bar{y}_{1:T}^i, e^i)q_\alpha(e^i|\bar{y}_{1:T}^i)}$. As described in Sec. 2, one of the challenges with integrating recordings in a common latent space is different dimensionalities (number of recorded neurons) as well as the dependence of neural activity on the shared latent space. We address this by training additional read-in networks $\Omega_i : \mathbb{R}^{d_y^i} \to \mathbb{R}^{d_{\bar{y}}}$ for each dataset that map $y_t^i$ to an intermediate vector, which we denote by $\bar{y}_t^i \in \mathbb{R}^{d_{\bar{y}}}$. This read-in network ensures that the latent states and dynamical-embeddings inferred from each dataset are aligned to live in the same space (Vermani et al., 2024b).

While there are many choices for parameterizing the encoders, we follow the parameterization in (Krishnan et al., 2015) for simplicity[2], defined as follows,

$$\bar{y}_{b,1:T}^i = \Omega^i(y_{b,1:T}^i), \tag{15}$$

$$q_\alpha(e^i \,|\, \bar{y}_{b,1:T}^i) = \mathcal{N}(e_b^i \,|\, \mathrm{agg}[\mu_\alpha(\bar{y}_{b,1:T}^i)], \mathrm{agg}[\sigma_\alpha^2(\bar{y}_{b,1:T}^i)]), \tag{16}$$

$$q_\beta(z_{1:T}^i \,|\, \bar{y}_{1:T}^i, e_b^i) = \prod_{t=1}^T \mathcal{N}(z_t^i \,|\, \mu_\beta(\mathrm{concat}[\bar{y}_{b,1:T}^i, e_b^i]), \sigma_\beta^2(\mathrm{concat}[\bar{y}_{b,1:T}^i, e_b^i])), \tag{17}$$

where $y_b^i$ denotes a randomly sampled mini-batch of trials $b$ from dataset $i$, concat is the concatenation operation, and agg is an aggregation operation. We aggregate the dynamical embedding over trials in a mini-batch that belong to the same dataset since we are interested in capturing inter-dataset, rather than intra-dataset variations, in the underlying dynamical systems. In practice, we parameterize $\mu_\alpha(\cdot)$, $\sigma_\alpha^2(\cdot)$ by a bidirectional recurrent neural network, and $\mu_\beta(\cdot)$, $\sigma_\beta^2(\cdot)$ by a regular recurrent network, and agg corresponds to a simple averaging function. We emphasize that $\mu_\alpha$, $\sigma_\alpha^2$, $\mu_\beta$, and $\sigma_\beta^2$ are shared across all datasets (See Fig. 14 for details on inference).

### 3.3 PROOF OF CONCEPT

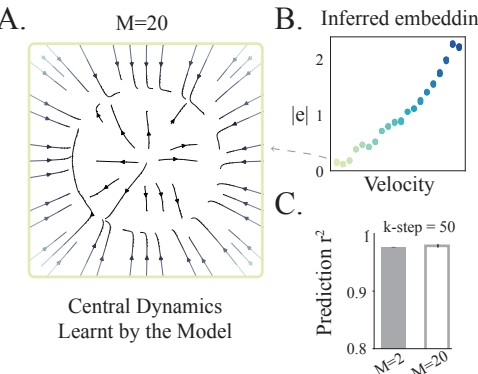

Figure 3: **A.** Mean dynamical system corresponding to the slowest velocity recording learned by the proposed approach when trained with $M = 20$ datasets. **B.** Samples from the inferred dynamical embedding for each dataset (see eq. 16). **C.** Forecasting $r^2$ at $(k = 50)$-step for models trained with $M = 2$ or $M = 20$ datasets.

As a proof of concept, we revisit the motivating example presented in Section 2 as a means to validate the efficacy of our approach and investigate how it unifies dynamics across datasets. For both $M = 2$ and $M = 20$ datasets, we used an embedding dimensionality of 1 and allowed the network to make a rank-1 change to the dynamics parameters.

After training, we observed that the shared dynamics (when $e = 0$) converged to a limit cycle with a slow velocity (Fig. 3A)—capturing the global topology that is shared across all datasets—and the model learned to modulate the velocity of the dynamics conditioned on the dynamical embedding which strongly correlated with the dataset specific velocity[3] (Fig 3B). This demonstrated that the proposed approach is able to capture dataset-specific variability. Lastly, Fig. 3C demonstrates that the proposed approach is able to forecast well for both $M = 2$ and $M = 20$ datasets. We include further validation experiments when there is no model mismatch as well as the generalization of the trained model to new data in Appendix B. We additionally include results on these recordings from multi-session CEBRA (Schneider et al., 2023) in Appendix B.

## 4 RELATED WORKS

**Multi-Dataset Training in Neuroscience**. Previous work has explored multi-dataset training for extracting latent representations in neuroscience, especially across datasets recorded during the same behavioral tasks. LFADS (Pandarinath et al., 2018), a variant of the seqVAE framework, used session-stitching with dataset-specific likelihood functions, but focused on single-animal recordings. Linderman et al. (2019) used a hierarchical Bayesian state-space model with switching linear dynamical systems, while Herrero-Vidal et al. (2021) developed a joint model with shared linear dynamics and dataset-specific likelihoods. In contrast to these approaches, we incorporate a more

---

[2]We evaluate alternative inference and learning formulations in Appendix D

[3]Note that we plot the absolute embedding samples since the likelihood function can introduce arbitrary invariance such as direction flipping, rotation, and so on.

expressive function to approximate the underlying family of dynamical systems which can disentangle variabilities across recordings. CEBRA (Schneider et al., 2023) and CS-VAE (Yi et al., 2023) have been developed for extracting latent representations by integrating multiple datasets. The multi-session training objective in CEBRA promotes invariant feature learning across datasets, while CS-VAE partitions the latent space to learn structured features from behavioral videos. In our framework, we jointly infer latent trajectories from multi-session recordings and learn a unified generative model that captures variations in dynamical systems from the inferred trajectories. Recently, there has been growing interest in using diverse neural recordings for training large-scale foundation models in neuroscience (Ye et al., 2023; Zhang et al., 2023; Caro et al., 2024; Azabou et al., 2024; Vermani et al., 2024a). These models leverage transformer-based architectures which lack recurrent hidden states and only incorporate temporal information indirectly via positional encoding. While our approach shares the same broad goal of pretraining a single generative model for rapid learning on downstream recordings, the focus of our work is on learning a family of dynamical systems underlying recordings.

**Recurrent Neural Network Models in Neuroscience** Integrative modeling of dynamical behaviors has also been explored in RNN models of neural systems (Yang et al., 2019; Driscoll et al., 2024). In Driscoll et al. (2024), the authors trained an RNN to perform multiple cognitive tasks and observed motifs corresponding to distinct dynamical behaviors. The broad idea of dynamical structure re-use is similar to our work but there are subtle differences—we are interested in capturing both topological and geometrical differences, and the "context" is learned from data. The motifs in Driscoll et al. (2024) corresponded to a distinct fixed point structure or topology with a pre-specified context input that could push the dynamics to task-relevant regions in the state space. The embedding analysis in Cotler et al. (2023), where a meta-model was trained to capture the activity of multiple trained RNNs is quite similar to our main idea since they observed similar dynamical properties in models that were close in the embedding space. Recent work on modeling motor adaptation (Pellegrino et al., 2023) by low-tensor rank learning in RNNs is broadly similar to our work since the authors adapt the weights in a low-rank RNN to capture variations in dynamics across trials. In contrast, we are interested in modeling dynamical variations across recording sessions and/or tasks.

Additional related works can be found in Appendix A.

## 5 EXPERIMENTS

We first validate the proposed method on synthetic data and then test our method on neural recordings from the primary motor and premotor cortex. We compare the proposed approach against the following baselines for all experiments.

We train a separate **Single Session** model using the seqVAE framework on each dataset. Given sufficient training data, this should result in the best performance, but will fail in trial-limited regimes. We consider a multi-session **Shared Dynamics** model with dataset-specific likelihoods (Pandarinath et al., 2018; Herrero-Vidal et al., 2021). We also compare against a baseline where the embedding is provided as an additional input to the dynamics model (**Embedding-Input**), a similar formulation to CAVIA (Concat) (Zintgraf et al., 2019) and DYNAMO (Cotler et al., 2023). We also test the hypernetwork parametrization proposed in CoDA (Kirchmeyer et al., 2022), where the hypernetwork adapts all parameters as a linear function of the dynamical embedding (**Linear-Adapter**).

We include additional baselines for the motor cortex experiment; we evaluate single session **LFADS** (Pandarinath et al., 2018) with the controller as an alternative generative model comparison. We also consider different methods for learning and inference. Specifically, we include single-session models as well as our proposed generative model trained using **Variational Sequential Monte Carlo** (VSMC) (Naesseth et al., 2018), and the **Deep Variational Bayes Filter** (DVBF) (Karl et al., 2016).

For all experiments, we split each of the $M$ datasets into a training and test set and report reconstruction and forecasting metrics on the test set. To measure the generalization performance, we also report these metrics on held-out datasets. Further details on training and evaluation metrics can be found in Appendix G.

## 5.1  BIFURCATING SYSTEMS

In these experiments, we test whether our method could capture variations across multiple datasets, particularly in the presence of significant dynamical shifts, such as bifurcations commonly observed in real neural populations. We selected two parametric classes of dynamical systems for testing: i) a system undergoing a Hopf bifurication and, ii) the unforced Duffing system. We include the results for training on datasets generated only from the Hopf system in Appendix C and we discuss the results from joint training on both systems here. We briefly outline the data generation process for the Duffing system (details of the data generation for the Hopf system can be found in Appendix E.2).

The latent trajectories for the Duffing system were generated from a family of stochastic differential equations,

$$\dot{z}_1 = z_2 + 5\,\mathrm{d}W_t, \qquad \dot{z}_2 = a^i z_2 - z_1(b^i + c z_1^2) + 5\,\mathrm{d}W_t \tag{18}$$

with $c = 0.1$, $a, b \in \mathbb{R}$, and $\mathrm{d}W_t$ denoting the Wiener process. In Fig. 4A, we visualize how the dynamical system changes as $a$ and $b$ vary. We chose $M = 20$ pairs of $(a^i, b^i)$ values (Fig 13),

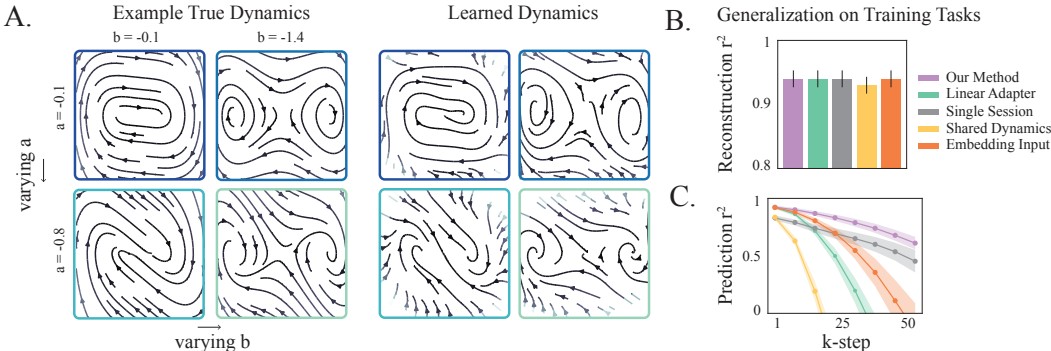

Figure 4: **A.** (Left) True underlying dynamics from some example datasets used for pretraining as a function of parameters $a$ and $b$ and (Right) the embedding conditioned dynamics learnt by our model. **B.**, **C.** Mean reconstruction and forecasting $r^2$ of the observations for all datasets used for pretraining on test trials.

and generated latent trajectories of length $T = 300$. Observations were generated according to $y_t^i \sim \mathcal{N}(C^i z_t^i, 0.01\mathbb{I})$ with the dimensionality of the observations varying between 30 and 100. In addition to these 20 Duffing system datasets, we included 11 datasets from the Hopf system (Appendix C), and used 128 trajectories from each of these 31 datasets for training all methods. We report performance on 64 test trajectories from each dataset. We used $d_e = 2$ for all embedding-conditioned approaches and constrained the hypernetwork to make rank $d_r = 1$ changes for our approach.

Our approach learned a good approximation to the ground-truth dynamics of the Duffing oscillator system, successfully disentangling different dynamical regimes (Fig. 4 B). Apart from learning the underlying topology of dynamics, it also better captured the geometrical properties compared to other embedding-conditioned baselines (Fig. 15). We observed similar results for datasets from the Hopf system–while our approach approximated the ground-truth system well, the Embedding-Input baseline displayed interference between dynamics and the Linear-Adapter learned a poor approximation to the ground-truth system (Fig. 16). Consequently, our approach outperformed other methods on forecasting observations with all methods having comparable reconstruction performance (Fig. 4B, C). Notably, apart from the $d_e$, we used the same architecture as when training on only the Hopf datasets, and did not observe any drop in performance for our approach, in contrast to baselines (Fig. 11C (Bottom), Fig. 4C).

Next, we tested the few-shot performance of all methods on new datasets, two generated from the Duffing oscillator system and one from the Hopf system, as a function of $n_s$, the number of trials used for learning the dataset specific read-in network, $\Omega^i$ and likelihood. Our approach and the Linear-Adapter demonstrated comparable forecasting performance when using $n_s = 1$ and $n_s = 8$ training trajectories. However, with $n_s = 16$ training trials, unlike other methods, our approach continued to improved and outperformed them (Table 1). This could be explained by looking at the

| | $n_s = 1$ | $n_s = 8$ | $n_s = 16$ |
|---|---|---|---|
| Our Method | **0.69 ± 0.072** | **0.78 ± 0.051** | **0.87 ± 0.037** |
| Linear-Adapter | **0.68 ± 0.08** | **0.79 ± 0.026** | 0.74 ± 0.039 |
| Single Session | 0.47 ± 0.119 | **0.79 ± 0.014** | 0.79 ± 0.047 |
| Shared Dynamics | -0.31 ± 0.103 | -0.34 ± 0.086 | -0.13 ± 0.065 |
| Embedding-Input | 0.59 ± 0.084 | **0.77 ± 0.04** | 0.74 ± 0.039 |

Table 1: Few shot forecasting performance ($k = 30$-step) on 3 held-out datasets as a function of $n_s$, the number of trials used to learn dataset specific read-in network and likelihood. (± 1 s.e.m)

inferred embedding on held-out datasets—as we increased the number of training trajectories, the model was able to consistently align to the "correct" embedding (Fig. 17).

## 5.2 MOTOR CORTEX RECORDINGS

Next, we tested the applicability of the proposed approach on neural data. We used single and multi-unit neural population recordings from the motor and premotor cortex during two behavioral tasks–the Centre-Out (CO) and Maze reaching tasks (Perich et al., 2018; Gallego et al., 2020; Churchland et al., 2012). In the CO task, subjects are trained to use a manipulandum to reach one of eight target locations on a screen. In the Maze task, subjects use a touch screen to reach a target location, while potentially avoiding obstacles. These recordings spanned different sessions, animals, and labs, and involved different behavioral modalities, while still having related behavioral components, making them a good testbed for evaluating various methods. For training, we used $40$ sessions from the CO task, from subjects M and C, and $4$ sessions from the Maze task from subjects Je and Ni. We set the dimensionality of latent dynamics to $d_z = 30$, and used an embedding dimensionality of $d_e = 2$, for all embedding-conditioned dynamics models. For our approach, we constrain the hypernetwork to make rank $d_r = 6$ changes, although we verified that the performance was not sensitive to $d_r$ (Fig 18). As a proxy for how well the various approaches learned the underlying dynamics, we report metrics on inferring the hand velocity using reconstructed and forecasted neural data from the models. Note that we align all recordings to the movement onset (details in Appendix G).

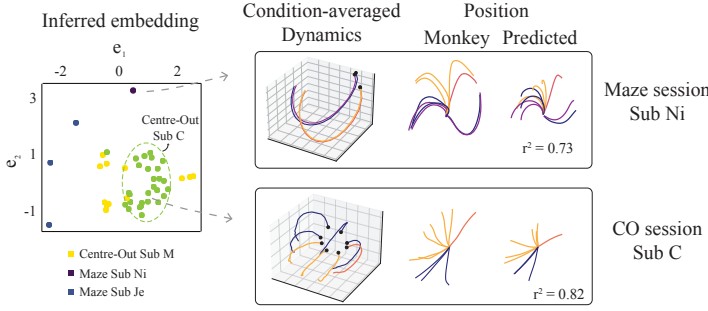

Figure 5: Visualizing the embedding manifold. (Left) Each point corresponds to a sample from the inferred embedding distribution (see eq. 16) corresponding to each recording. (Right) The condition-averaged latent dynamics for a session from Maze (Sub Ni) (Top) and a CO Session (Bottom) generated by the model, along with the corresponding real and forecasted behavior.

The inferred dynamical embedding displayed distinct structures across behavioral tasks and subjects (Fig. 5, Left). While the CO task involves more stereotyped straight reaching behavior with the same stimulus conditions across datasets, the Maze task has more complex stimulus statistics which vary across sessions. The family of learned dynamics reflected this heterogeneity across recordings. We visualize these learned dynamical systems for two example sessions, one from each task, in Fig 5 (Right). Specifically, we used the trained encoders, $q_\beta$ and $q_\alpha$ to estimate the latent state and embedding at the beginning of movement onset. We subsequently generate the latent dynamics from that state using $f_{\theta,e^i}$ till the end of the movement onset. The condition-averaged principal components (PCs) of these generated latents are shown in the figure.

We observed that most of the approaches had adequate performance on reconstructing velocity from neural recordings, with our method and Linear-Adapter outperforming single session reconstruction performance on the CO task (Fig. 6A, top). Multi-Session CEBRA was not able to adequately capture the variability in the Maze sessions and had low reconstruction $r^2$. In terms of forecasting, the single-session model trained using the seqVAE framework had the best performance. Notably, our approach managed to balance learning both the CO and Maze tasks relative to other multi-session baselines, with all performing better on the CO task than the Maze (Fig. 6A, bottom). The generative model learned from CEBRA had poor forecasting performance which resulted in a negative $r^2$ value (not plotted). Next, we tested if we can transfer these learned dynamics to new recordings as we varied $n_s$ from 8 to 64 trials for learning the read-in network and likelihood. We used trials from 2 held-out sessions from Sub C and M, as well as 2 sessions from a new subject (Sub T) for evaluating all methods. We observed that our approach consistently performed well on both reconstruction and forecasting for held-out sessions from previously seen subjects, and reached good performance on sessions from Sub T as we increased the training trials (Fig. 6B, C ($n_s = 32$)). Moreover, our method outperformed all other baselines on forecasting, especially in very low-sample regimes, while having comparable reconstruction performance (Fig. 19).

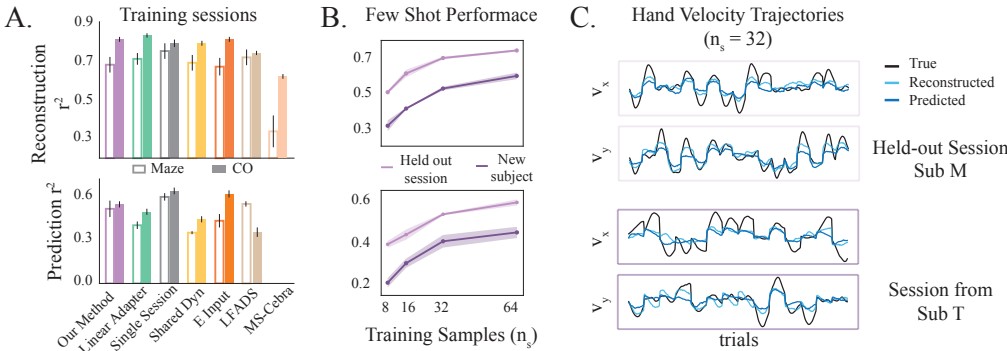

Figure 6: **A.** (top) $r^2$ for hand velocity decoding from reconstructed and (bottom) forecasted neural observations for Maze and Centre-Out sessions. **B.** Behavior reconstruction (top) and forecasting (bottom) performance on held-out sessions and sessions from a new subject as a function of the number of training samples. **C.** Hand velocity trajectories (400 ms after movement onset) predicted by our approach on 17 test trials from held-out session (top) and 13 test trials from a session on a new subject (bottom), after using $n_s = 32$ trials for aligning to the pre-trained model.

Next, we evaluated the impact of the inference framework on effective learning and few-shot performance. We specifically tested single session models as well as our proposed generative model trained after performing inference using VSMC and DVBF (Details in Appendix D). In both cases, we observed that the inferred embedding distribution learned the underlying dynamical structure across datasets (Fig. 12A). Moreover, we were able to similarly exploit this learned structure for few-shot forecasting on novel recording sessions (Fig. 12B). We additionally investigated the effect of large-scale training for sample-efficient transfer on downstream tasks by only pretraining the model on 128 trials from 4 sessions spanning different tasks and subjects. Even in this case, the embedding distribution displayed clear clustering based on the task and subject. Moreover, the model performed comparably to the Single-Session model on reconstruction, while outperforming it on prediction for both tasks (Fig. 20 A, B). However, it demonstrated poor performance on new sessions given limited trials for learning the read-in and likelihood parameters (Fig. 20 C), underscoring the importance of large-scale training for generalizing to novel settings.

Finally, we probed the differences in the latent state evolution given the same initial condition while interpolating across the learned embedding. In order to do this, we chose an example session from the Maze and CO datasets and obtained their corresponding dynamical embedding from the model, shown as a solid blue and green circle in Fig. 7 (middle), respectively. A grid of points was sampled around each of these inferred embeddings (shown as shaded squares in Fig. 7 middle), and for each point we obtained the corresponding low-rank parameter changes to generate the latent trajectories. We observed that the embedding space learned a continuous representation of dynamics, which was reflected in similar predicted behaviors close to the original learned embedding (Fig 7). Interestingly,

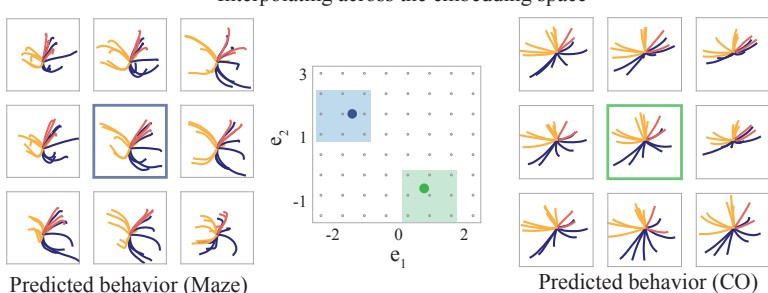

Figure 7: The predicted behavior for a Maze (Sub Je) session and CO (Sub C) session at 9 grid points around the original inferred embedding. The point closest to the original embedding is highlighted in blue and green respectively.

when we interpolated through the entire embedding space, the predicted behavior and corresponding dynamics continuously varied as well. Specifically, the predicted behavior and dynamics trajectories on the CO session demonstrated similarities over a large portion of the embedding space, with the trajectories shifting to more curved reaches further from the original embedding (Fig. 21). On the Maze task, the trajectories demonstrated more heterogeneity in responses, and decayed to a fixed point further away from the original embedding (Fig. 22).

## 6  DISCUSSION

We present a novel framework for jointly inferring and learning latent dynamics from heterogeneous neural recordings across sessions/subjects during related behavioral tasks. To the best of our knowledge, this is the first approach that facilitates learning a family of dynamical systems from heterogeneous recordings in a unified latent space, while providing a concise, interpretable manifold over dynamical systems. Our meta-learning approach mitigates the challenges of statistical inference from limited data, a common issue arising from the high flexibility of models used to approximate latent dynamics. Empirical evaluations demonstrate that the learned embedding manifold provides a useful inductive bias for learning from limited samples, with our proposed parametrization offering greater flexibility in capturing diverse dynamics while minimizing interference. We demonstrate that the few-shot performance of our proposed generative model is largely agnostic to the inference method. We observe that the generalization of our model depends on the amount of training data—when trained on smaller datasets, the model learns specialized solutions, whereas more data allows it to learn shared dynamical structures. This work enhances our capability to integrate, analyze, and interpret complex neural dynamics across diverse experimental conditions, broadening the scope of scientific inquiries possible in neuroscience.

### LIMITATIONS AND FUTURE WORK

Our current framework uses event aligned neural observations; in the future, it would be useful to incorporate task-related events, to broaden its applicability to complex, unstructured tasks. Further, the model's generalization to novel settings depends on accurate embedding inference, a challenge noted in previous works that disentangle task inference and representation learning (Hummos et al., 2024). However, we observe consistent improvement in embedding inference with increase in the number of training samples from novel recordings. Our empirical observations demonstrate that using a hypernetwork improves the expressivity of the dynamical systems model relative to other parametrizations. It would be interesting to investigate the theoretical basis of this observation in the future. While our latent dynamics parametrization is expressive, it assumes shared structure across related tasks. Future work could extend the model to accommodate recordings without expected shared structures (for instance, by adding explicit modularity (Márton et al., 2021)). Investigating the performance of embedding-conditioned low-rank adaptation on RNN-based architectures presents another avenue for future research. Finally, the embedding manifold provides a map for interpolating across different dynamics. While we focus on rapid learning in this paper, our framework could have interesting applications for studying inter-subject variability, learning-induced changes in dynamics, or changes in dynamics across tasks in the future.

ACKNOWLEDGMENTS

We would like to thank the anonymous reviewers for providing helpful feedback on the manuscript. This work was supported by NIH RF1-DA056404 and the Portuguese Recovery and Resilience Plan (PPR), through project number 62, Center for Responsible AI, and the Portuguese national funds, through FCT - Fundação para a Ciência e a Tecnologia - in the context of the project UIDB/04443/2020.

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

## CONTENTS

## A   ADDITIONAL RELATED WORKS

Several meta-learning approaches have been developed for few-shot adaptation, including gradient-based methods (Finn et al., 2017; Li et al., 2017; Nichol & Schulman, 2018; Zintgraf et al., 2019). Amongst these, LEO (Rusu et al., 2019) shares the same idea of meta-learning in low-dimensional space of parameter embeddings. However, gradient-based approaches require fine-tuning during test-time, and have had limited success for meta-learning dynamics. Similar to our work (Cotler et al., 2023) also learns an embedding space of dynamics learned from trained RNNs, however, we are interested in learning dynamics directly from data. Some methods for learning generalizable dynamics been previously proposed—DyAD (Wang et al., 2022) adapts across environments by neural style transfer, however it operates on images of dynamical systems, LEADS (Yin et al., 2021) learns a constrained dynamics function that is directly added to some base dynamics function, and CoDA (Kirchmeyer et al., 2022) which learns task-specific parameter changes conditioned on a low-dimensional context similar to our approach. However, these approaches were applied in supervised settings on low-dimensional systems whereas we operate in an unsupervised setting.

## B   PROOF-OF-CONCEPT EXPERIMENT

**1-shot Performance**. We evaluated the generalization performance of our approach on a new dataset with $\omega^{M+1} = 4.1$, which was not included in the training set, by using 1 training trajectory to train a new read-in network, $\Omega^{M+1}$ and likelihood $p_\phi^{M+1}$. After training, the model displayed similar prediction performance on the new dataset ($r_{k=50}^2 = 0.94 \pm 0.001$) (Fig. 8).

**No Model Mismatch**. Here, we investigated the performance of our approach when there is no mismatch between the proposed generative model and the true system. For this experiment, we generated synthetic data from the model trained on $M = 20$ datasets. We used the observations from validation trials till $t = 100$ to infer the embedding, $e^i$, and latent state, $z_t^i$. We subsequently used the dynamics model to generate latent trajectories of length 250, $z_{t+1:t+250}^i$ and mapped them back to the observations via the learned

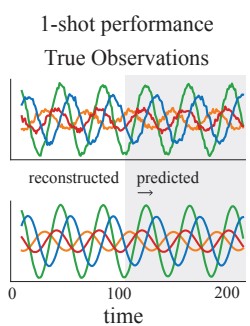

Figure 8: (Top) True observations on new data with $\omega = 4.1$ and (Bottom) the corresponding reconstructed and predicted observations after aligning to the trained model.

likelihood function. We re-trained a model with the same architecture while keeping the likelihood readout and read-in parameters fixed since the likelihood could arbitrarily flip the direction of the flow field.

Similar to the ground truth generative model, the inferred embedding co-varied with the different velocities (Fig. 9A, left). Further, the model recovered the correct topology of the ground truth dynamics (Fig. 9B), reflected in the forecasting performance on held out test trials (Fig. 9A, right).

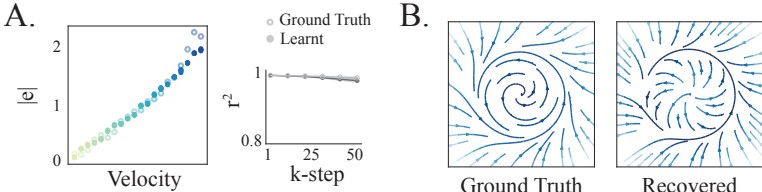

Figure 9: **A.** (Left) Samples from the inferred embedding (see eq. 16) after training on trajectories generated from our model overlaid on the ground truth embeddings. (Right) Forecasting performance of the trained and ground truth model on held out trials. **B.** Ground truth dynamics generated from an embedding sample and the corresponding recovered dynamics.

**Multi-Session CEBRA.** We evaluate the performance of multi-session CEBRA, an approach for inferring latents by integrating datasets. This variant of CEBRA is designed to learn invariant latent features across datasets, and has not been evaluated on recordings with variations in underlying dynamical features. In this experiment, we fit $M = 20$ datasets using CEBRA and post-hoc trained a generative model with shared dynamics and dataset-specific likelihood functions, since it does not learn a generative model. After training CEBRA on these datasets, we observed that the model recovered oscillatory latent trajectories; however, these trajectories were jagged and did not capture the characteristics of the true latents (Fig. 10A). Next, we trained a generative model using these latent trajectories. We observed that the learned dynamical system managed to capture a global limit-cycle like structure (Fig. 10B, left). However, this limit cycle also contained a fixed-point like structure causing rapid slow-down or noise-induced oscillations, capturing the characteristics of the latent trajectories inferred by CEBRA. Due to this behavior, we observed poor forecasting and reconstruction of observations (Fig. 10B, right).

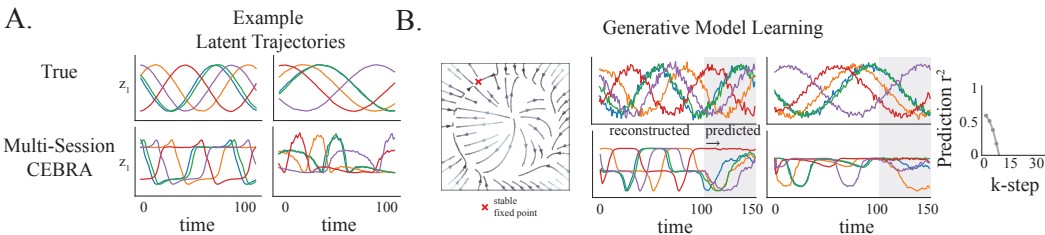

Figure 10: **A.** Example latent trajectories from two datasets (Top) and inferred latent trajectories from Multi-Session CEBRA trained on $M = 20$ datasets. **B.** The dynamics learned by a generative model trained on the latent trajectories (Left). Example reconstructed and predicted observations on the two datasets by using the learned generative model (Middle). The k-step prediction $r^2$ of the forecasted observations for $M = 20$ datasets.

## C    HOPF BIFURCATION SYSTEMS

For all embedding conditioned approaches, we set $d_e = 1$ and learned a rank-1 change to the dynamics for our approach. Our approach successfully learned both dynamical regimes present in the datasets and the embedding distribution encoded differences in these dynamics with high certainty given limited time bins on test trials(Fig. 11A, B). While all approaches performed well on reconstructing observations on these datasets, our approach and the Embedding-Input outperformed other multi-session baselines on forecasting (Fig 11C). We also evaluated the generalization performance of all methods on the 2 held-out datasets as a function of training data used for training the read-in network

and observed similar trends as demonstrated by the reconstruction and k-step $= 20 \, r^2$ on test trials from these datasets, shown in Table 2.

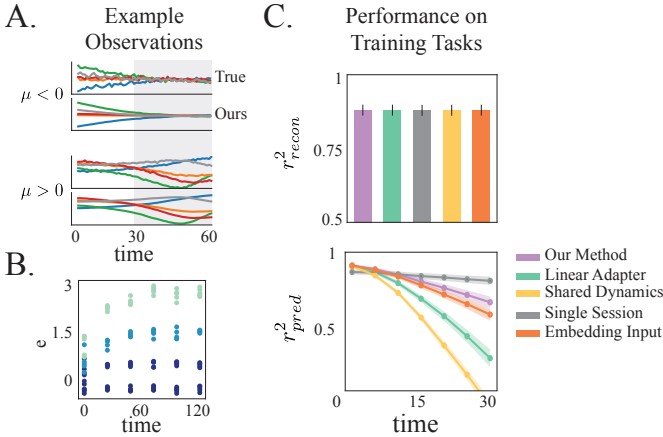

Figure 11: **A.** Example observations along with the reconstructed and predicted observations from our approach for the fixed point (Top) and limit cycle (Bottom) dynamical regimes. **B.** Samples from the embedding as a function of the number of time bins in the test trials for 4 example datasets. **C.** Reconstruction (Top) and forecasting (Bottom) performance of all approaches on the datasets used for pretraining.

| | Reconstruction | | Forecasting | |
| --- | --- | --- | --- | --- |
| | $n_s = 1$ | $n_s = 8$ | $n_s = 1$ | $n_s = 8$ |
| Ours | $0.85 \pm 0.054$ | $0.89 \pm 0.04$ | $0.64 \pm 0.1$ | $0.69 \pm 0.07$ |
| Linear-Adapter | $0.84 \pm 0.059$ | $0.89 \pm 0.04$ | $-0.1 \pm 0.34$ | $0.55 \pm 0.08$ |
| Single Session | $0.8 \pm 0.054$ | $0.88 \pm 0.044$ | $0.27 \pm 0.08$ | $0.77 \pm 0.03$ |
| Shared Dynamics | $0.83 \pm 0.068$ | $0.89 \pm 0.04$ | $0.32 \pm 0.08$ | $0.32 \pm 0.04$ |
| Embedding-Input | $0.86 \pm 0.049$ | $0.89 \pm 0.04$ | $0.61 \pm 0.09$ | $0.56 \pm 0.11$ |

Table 2: Few-shot reconstruction and forecasting performance (k-step=20) for held-out datasets in C where $n_s$ is the number of trials used for learning the dataset specific read-in network and likelihood.

# D  ALTERNATIVE INFERENCE AND LEARNING APPROACHES

While we focus on the DKF method for performing inference in the main text, we evaluate the efficacy of our proposed generative model with alternative inference and learning schemes.

## D.1  VARIATIONAL SMC

We considered the variational sequential monte carlo (VSMC) framework proposed by Naesseth et al. (2018) for learning and inference. VSMC is a combination of variational inference and sequential Monte Carlo (SMC) (Doucet et al., 2001), allowing for optimization of the parameters of the proposal; for simplicitiy, unlike traditional SMC, no resampling was performed after every time step, $t$. Given $N$ samples from the encoder, i.e., $z_{1:T}^1, \ldots, z_{1:T}^N$, VSMC optimizes the following lower bound to the log marginal likelihood,

$$\log p(y_{1:T}) = \log \int \prod_{t=1}^{T} p(z_t \,|\, z_{t-1}) p(y_t \,|\, z_t) dz_{1:T} \geq \tilde{L}_{vsmc} = \sum_{t=1}^{T} \mathbb{E}_{q(z_t)} \left[ \log \left( \frac{1}{N} \sum_{i=1}^{N} w_t^i \right) \right],$$

where,

$$w_t^i = \frac{p(y_t \,|\, z_t^i) p(z_t^i \,|\, z_{t-1}^i)}{q(z_t^i \,|\, y_{1:T})}. \tag{20}$$

As the proposed formulation requires inference of the dynamical embedding, $e$—which is constant over time—we have to modify VSMC as it non-trivial to infer constants in state-space models using

SMC (Doucet et al., 2001). We can express the log marginal likelihood of the proposed generative model as

$$\log p(y_{1:T}) = \log \int p(e) \prod_{t=1}^{T} p(z_t \,|\, z_{t-1}, e) p(y_t \,|\, z_t) dz_{1:T} de, \tag{21}$$

$$= \log \int p(e) de \int \prod_{t=1}^{T} p(z_t \,|\, z_{t-1}, e) p(y_t \,|\, z_t) dz_{1:T}, \tag{22}$$

$$= \log \int p(e) p(y_{1:T} \,|\, e) de, \tag{23}$$

$$= \log \int q(e) \frac{p(e)}{q(e)} p(y_{1:T} \,|\, e) de, \tag{24}$$

$$= \log \mathbb{E}_{q(e)} \left[ \frac{p(e)}{q(e)} p(y_{1:T} \,|\, e) \right]. \tag{25}$$

Applying Jensen's inequality to (25),

$$\log p(y_{1:T}) \geq \mathbb{E}_{q(e)} \left[ \log \left( \frac{p(e)}{q(e)} p(y_{1:T} \,|\, e) \right) \right], \tag{26}$$

$$\log p(y_{1:T}) \geq \mathbb{E}_{q(e)} \left[ \log p(y_{1:T} \,|\, e) \right] + \mathbb{E}_{q(e)} \left[ \log p(e) \right] - \mathbb{E}_{q(e)} \left[ \log q(e) \right]. \tag{27}$$

As expectations respect inequalities, we can lower bound $\mathbb{E}_q \left[ \log p(y_{1:T} \,|\, e) \right]$ using the VSMC lower-bound (19), leading to the following lower-bound that we optimize

$$\log p(y_{1:T}) \geq \mathbb{E}_{q(e)} \left[ \tilde{L}_{vsmc} \right] + \mathbb{E}_{q(e)} \left[ \log p(e) \right] - \mathbb{E}_{q(e)} \left[ \log q(e) \right], \tag{28}$$

$$\log p(y_{1:T}) \geq \sum_{t=1}^{T} \mathbb{E}_{q(z_t|e)q(e)} \left[ \log \left( \frac{1}{N} \sum_{i=1}^{N} w_t^i(e) \right) \right] + \mathbb{E}_{q(e)} \left[ \log p(e) \right] - \mathbb{E}_{q(e)} \left[ \log q(e) \right], \tag{29}$$

where

$$w_t^i(e) = \frac{p(y_t \,|\, z_t^i) p(z_t^i \,|\, z_{t-1}^i, e)}{q(z_t^i \,|\, y_{1:T}, e)}. \tag{30}$$

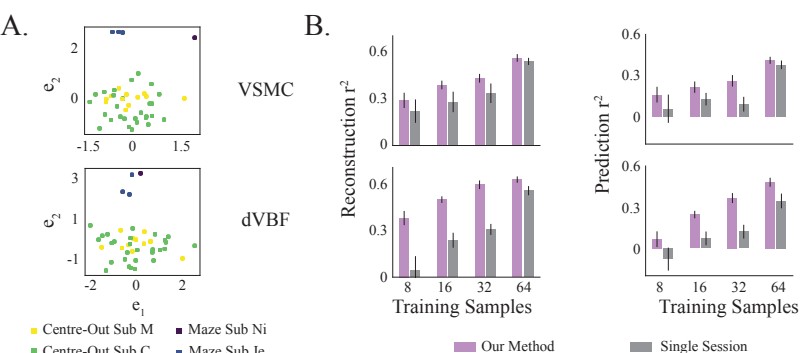

Figure 12: **A.** Samples from the learned embedding distribution using VSMC (Top) and dVBF (Bottom). **B.** Behavior decoding performance from reconstructed (Left) and forecasted trajectories (Right) using the two inference methods with a single session generative model vs aligning to our pretrained generative model.

## D.2 DEEP VARIATIONAL BAYES FILTER

We additionally considered the DVBF framework proposed in (Karl et al., 2016) for performing learning and inference. This framework explicitly ties the inference network to the generative model by forcing the samples from the inference network through the dynamical systems model. In our implementation, we defined the inference network as follows,

$$q_\beta(u_t^i \,|\, \bar{y}_{1:T}^i, e_b^i) = \mathcal{N}(u_t^i \,|\, \mu_\beta(\mathrm{concat}[\bar{y}_{b,t:T}^i, e_b^i]), \sigma_\beta^2(\mathrm{concat}[\bar{y}_{b,t:T}^i, e_b^i])),$$

where $q_\beta$ encoded the observations backward in time and was parametrized by an RNN to infer the parameters of the Gaussian distribution, $u_t \sim q(u_t)$. The latent trajectory for each dataset was subsequently sampled as,

$$z_t^i = f_{\theta, e^i}(z_{t-1}^i) + Q^{1/2} u_t^i.$$

Parameters of the generative model and inference networks were learned jointly by optimizing the following ELBO,

$$\mathcal{L} = \sum_i \sum_t \mathbb{E}_{q_{\alpha,\beta}}[\log p(y_t^i | z_t^i)] - \mathbb{E}_{q_\beta}[\mathbb{D}_{KL}(q(u_t)||p(u_t))] - [\mathbb{D}_{KL}(q_\alpha(q^i)||p(e))], \qquad (31)$$

where $p(u_t) \sim \mathcal{N}(0, \mathbf{I})$.

## E    DATA GENERATION DETAILS

### E.1    LIMIT CYCLE

For the experiments in Sections 2, 3.3, we simulated data from the following system of equations,

$$\dot{r} = r(1 - r)^2,$$
$$\dot{\theta} = \omega^i,$$
$$z_1^i = r \cos\theta + 5\, dW_t, \quad z_2^i = r \sin\theta + 5\, dW_t,$$

where $\omega^i, i \in \{1, \cdots, M\}$ is the dataset specific velocity and $dW_t$ is the Wiener process. Specifically, for the experiment with $M = 2$ datasets, we set $\omega^1 = 2$ and $\omega^2 = 5$; for the experiment with $M = 20$ datasets, we uniformly sampled 20 values for $\omega^i$ between 0.25 and 5. For each value of $\omega^i$, we generated 128 latent trajectories for training, 64 for validation and 64 for testing, each of length $T = 300$, where we used Euler-Mayurama with $\Delta t$ of 0.04 for integration. Observations were generated according to $y_t^i \sim \mathcal{N}(C^i z_t^i, R)$ where $R = 0.01 * I$ and the elements of the readout matrix $C^i$ were sampled from $\mathcal{N}(0, I/\sqrt{d^z})$; the dimensionality of the observations varied between 30 and 100. with $R \sim \mathcal{N}(0, 0.01)$, where the dimensionality of the observations varied between 30 and 100.

### E.2    HOPF BIFURCATION

Latent trajectories were generated from the following two-dimensional dynamical system,

$$\dot{z_1} = z_2 + 5\, dW_t, \qquad \dot{z_2} = -z_1 + (\mu^i - z_1^2)z_2 + 5\, dW_t, \qquad (32)$$

where the parameter $\mu^i$ controls the topology of the dynamics. Specifically, when $\mu^i < 0$, this system follows fixed point dynamics, and when $\mu > 0$ the system bifurcates to a limit cycle.

We uniformly sampled $M = 21$ values of $\mu^i$ between -1.5 and 1.5, and for each $\mu^i$, we generated 128 trajectories for training and 64 for testing, each trajectory was $T = 350$. Observations were generated according to $y_t^i \sim \mathcal{N}(C^i z_t^i, R)$ where $R = 0.01 * I$ and the elements of the readout matrix $C^i$ were sampled from $\mathcal{N}(0, I/\sqrt{d^z})$; the dimensionality of the observations varied between 30 and 100. For evaluating few-shot performance, we generated two additional novel datasets where $\mu^{M+1} = -0.675$ and $\mu^{M+2} = 1.125$ (both values were not included in the training set).

### E.3    DUFFING SYSTEM

For the Duffing system described in 18, we set $c = \frac{1}{10}$ and varied values of $a$ and $b$ (shown in blue, Fig. 13) for generating 20 datasets. We additionally used 11 datasets from C obtained by uniformly sampling $\mu$ between -1.5 and 1.5. For few shot evaluation of various approaches, we used two held-out datasets from the Duffing system (shown in orange, Fig. 13), as well as a dataset from the Hopf example generating by setting $\mu = -1.8$. All empirical evaluation was performed on 64 test trials from each dataset.

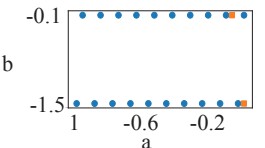

Figure 13: $(a^i, b^i)$ values used to generate different datasets from the Duffing system.

### E.4   MOTOR CORTEX RECORDINGS

We binned the spiking activity in 20ms bins and smoothed it with a 25ms causal Gaussian filter to obtain the rates for all datasets. We further removed neurons that had a firing rate of less than 0.1Hz and aligned the neural activity to movement onset. We used 512 trials when available or 80 percent of the trials, each of length 36, for training all methods.

## F   ADDITIONAL FIGURES

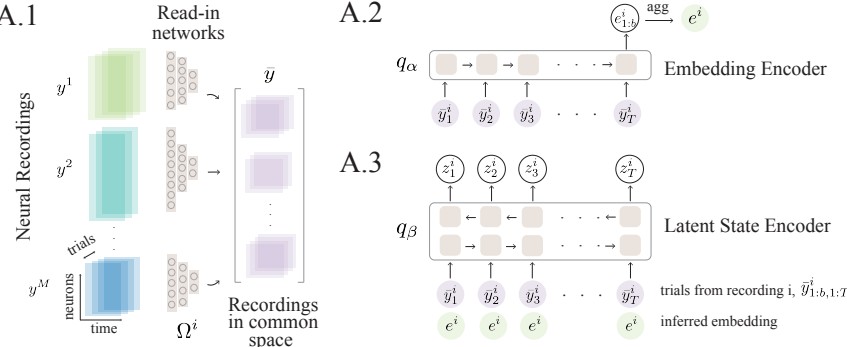

Figure 14: Inference Overview. **A.1**. Each recording $y^i_{1:T}$ is projected into a common space $\bar{y}^i_{1:T}$ by recording specific read-in networks $\Omega^i : \mathbb{R}^{d_{y^i}} \to \mathbb{R}^{d_{\bar{y}}}$. **A.2**. After being projected in this common space, the recording is processed by an RNN ($q_\alpha$) which infers the distribution over the dynamical embedding. Note that the dynamical embedding is aggregated across trials belonging to the same recording session. **A.3**. This inferred embedding is concatenated with $\bar{y}_{1:T}$ to obtain the latent state trajectories for each recording via the encoder $q_\beta$, parametrized by a bi-directional RNN.

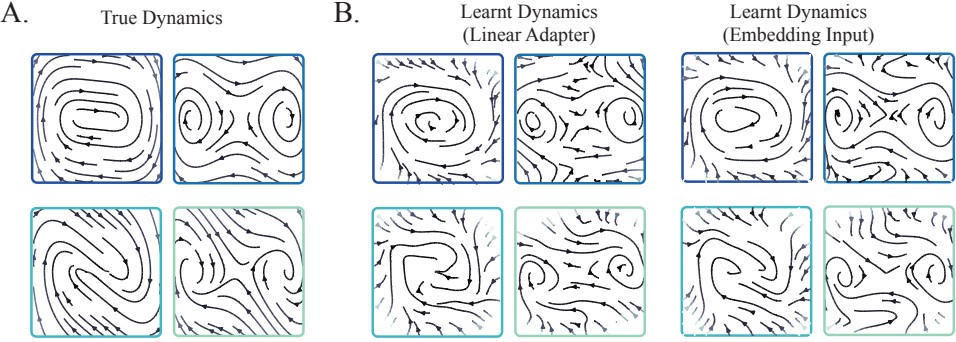

Figure 15: (Left) Dynamics learnt by Linear-Adapter and (Right) Embedding-Input corresponding to the example dynamics on the true system shown in Fig. 4A.

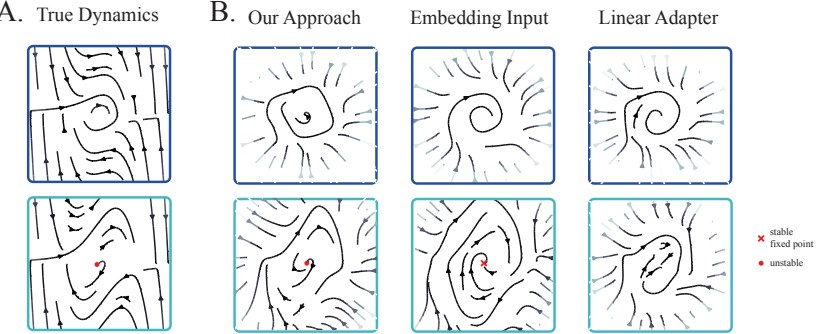

Figure 16: (Left) True dynamics from example datasets used for pretraining in experiment 5.1. (Right) Dynamics Learnt by different embedding-conditioned parametrizations.

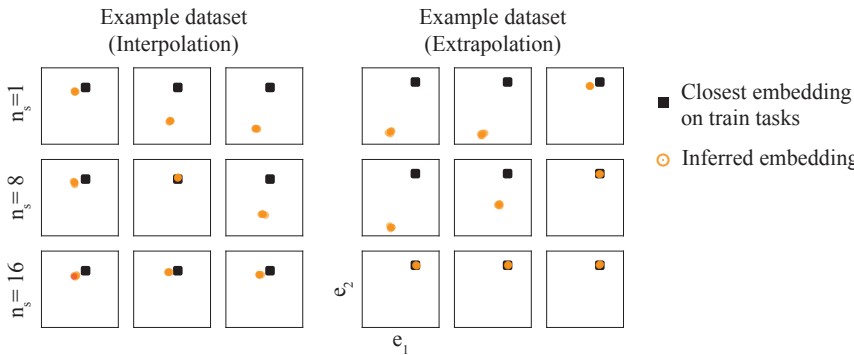

Figure 17: Samples from the embedding distribution as a function of the number of training trajectories on 3 seeds on held-out datasets from the Duffing system (denoted by orange, Fig. 13)

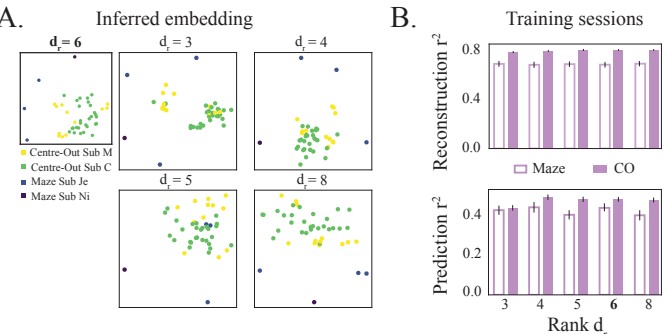

Figure 18: **A.** Sample from the inferred embedding for different ranks (best of 3 seeds).**B.** The behavior reconstruction (Top) and prediction (Bottom) $r^2$ for different ranks averaged over 3 training seeds.

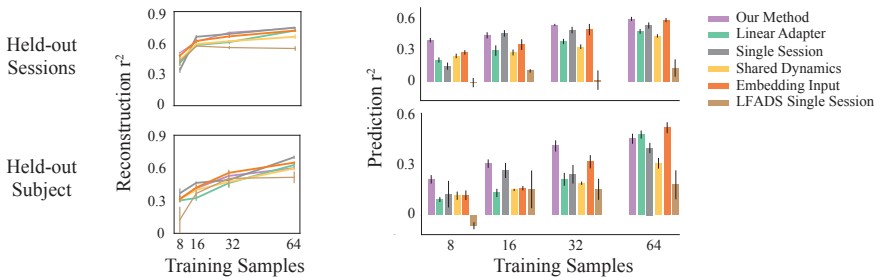

Figure 19: Behavior reconstruction (Left) and forecasting (Right) for all methods as a function of the number of training samples for held-out sessions from Sub M and C, and two sessions from a held-out Subject.

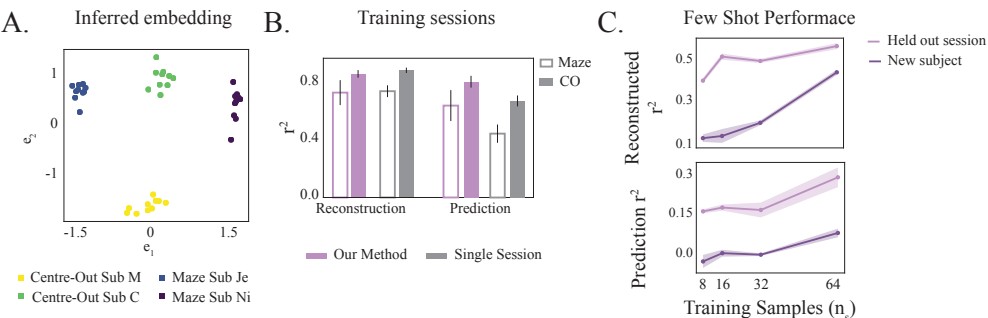

Figure 20: **A.** Samples from the embedding distribution after training our approach using 2 CO sessions from Sub C and M, and 2 Maze sessions from Sub Je and Ni. **B.** Reconstruction and forecasting performance of the model on held out test trials relative to the Single Session models. **C.** Few-shot reconstruction and forecasting performance on held out sessions and a new subject (Sub T).

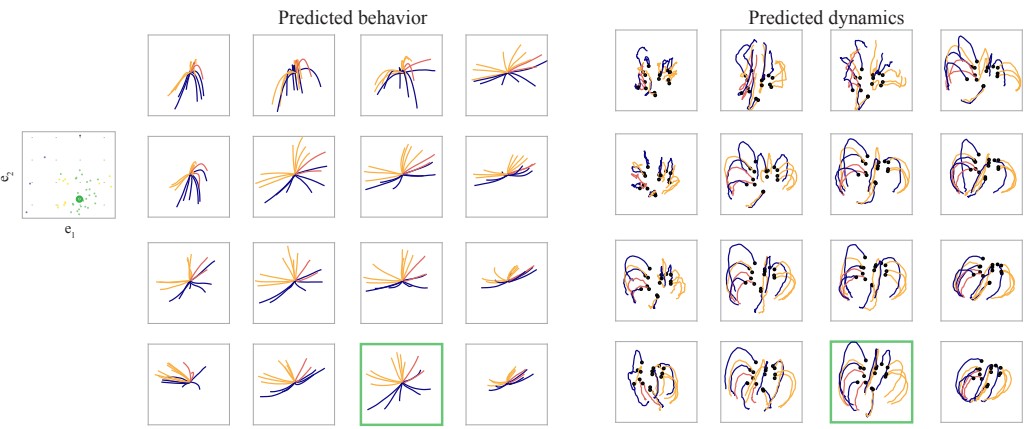

Figure 21: (Left) Grid points used for generating the latent dynamical trajectories and with the inferred embedding distribution overlaid. The embedding of the CO session from Sub C used to infer the initial condition of the latent state is highlighted in green. (Right) Single trials of the predicted hand position and PC projections of the corresponding latent dynamics trajectories. The initial latent state is denoted by the black points.

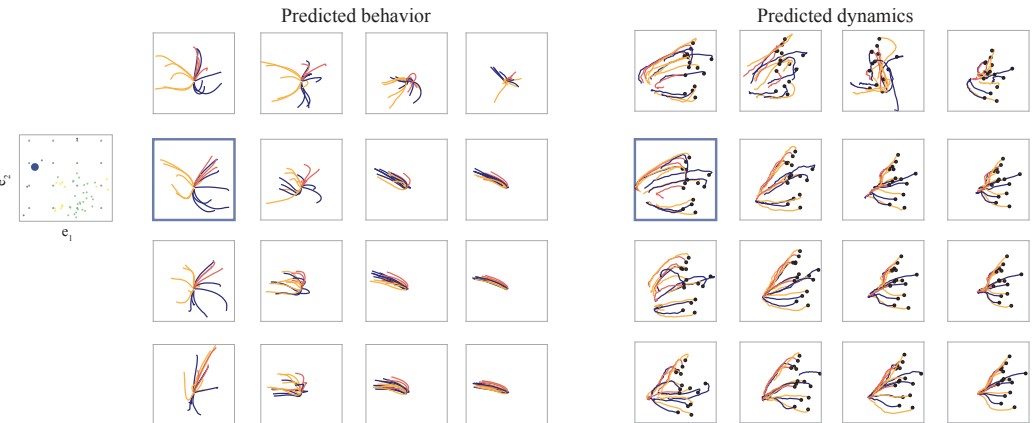

Figure 22: Same as 21 but for a Maze session from Sub Je (highlighted in blue).

# G    EXPERIMENT DETAILS

## G.1    TRAINING

Since CAVIA (Concat) (Zintgraf et al., 2019), DYNAMO (Cotler et al., 2023) and CoDA (Kirchmeyer et al., 2022) have not been developed for joint inference and learning of dynamics, we use our framework for inference with modifications to the parametrization of the embedding conditioned dynamics function on all experiments. We used the official implementation of LFADS in PyTorch to obtain the results reported in the paper (Sedler & Pandarinath, 2023).

For our method and CoDA, we restricted parameter changes to the input, $\mathbf{W}_{in}$, and hidden weights, $\mathbf{W}_{hh}$. We additionally included the Frobenius norm on the embedding-conditioned weight change $\|h_\varphi(e^i)\|_P$ along with the ELBO (eq. 14) for both approaches.

**Synthetic Experiments**. For the results reported in C and 5.1, we used the Adam optimizer with weight decay and a Cosine annealing schedule on the learning rate for pretraining all approaches.

**Motor Cortex Recordings**. We used the LAMB optimizer for pretraining all multi-session methods on the motor cortex recordings and used Adam with weight decay for the single-session models, with a Cosine annealing schedule on the learning rate in both cases. We also incorporated masking during training for all approaches to encourage learning better latent dynamics. Specifically, we sampled the latent state from the dynamics instead of the state inference network on randomly masked time bins to compute the likelihood.

**Aligning New Data**. We pretrained all multi-session approaches for 3 seeds and picked the best performing model to evaluate few-shot performance. When aligning new data to this pre-trained model, we trained the dataset-specific read-in network, $\Omega^i$ and likelihood functions $p_\phi^i$ for the new dataset, in addition to the state noise, $Q^i$, by optimizing the ELBO for new data (eq. 14). We used Adam with weight decay for aligning, and additionally incorporated masking when aligning held-out motor cortex datasets.

## G.2    FIGURE GENERATION

**Vector Field**. We generated all the vector field plots for synthetic experiments by sampling random points on a 2-D grid to obtain $z_t$. We used the mean dynamics learned by the model to estimate the velocity at $z_t$ as, $z_{t-1} = f_\theta(z_t) - z_t$. In the embedding-conditioned methods, we additionally used the inference network, $q_\alpha$ to estimate the dynamical embedding corresponding to each dataset, which was used to conditionally generate the vector field plots.

We additionally align all learned vector field plots to the true system for ease of visual comparison (Fig. 4A, 15, 16). We do do this by learning a linear transformation from the true latent trajectories to the latent trajectories learned by the model. Note that we follow the same procedure when visualizing latent trajectories inferred by multi-session CEBRA (Fig. 10A)

**Context Interpolation**. We fed the neural recordings up till movement onset time to the latent state encoder, $q_\beta$, to obtain the latent state at movement onset, $z_t$. We sampled points on a 2-D grid and simulated the corresponding samples from the embedding distribution as $e \sim \mathcal{N}(e, 0.1\mathbf{I})$. Given the latent state at movement onset for a particular recording session, we were able to obtain different dynamical trajectories by giving these embedding samples to the latent dynamics model $f_{\theta,e}(z_t)$ along with $z_t$. We used this procedure to obtain the results in in Fig. 7, 21 and 22.

## G.3    METRICS

**Synthetic Experiments**. We report the $r^2$ on observation reconstruction for test trials over the entire length of the trial for all approaches. In order to evaluate the forecasting performance, we use observations till time $t$ and sample the corresponding latent trajectories, $z_{0:t}^i$, from the inference network and the corresponding $e^i$ for the embedding-conditioned methods. We subsequently use the learned dynamics model to generate K steps ahead from $z_{t+1:t+K}^i$ and map these generated

trajectories back to the observations. The k-step $r^2$ for each dataset is computed as,

$$r_k^2 = 1 - \frac{\sum_{i=1}^{M}(y_k - \hat{y}_k)^2}{\sum_{i=1}^{M}(y_k - \bar{y})^2}$$

where $\bar{y}$ is the mean activity during the trial, and $M$ is the number of test trials.

**Motor Cortex Recordings**. On the motor cortex experiments, we report behavior decoding from the reconstructed and forecasted observations for all methods. Specifically, for each session, we trained a linear behavior decoder from the neural observations to the hand velocity of the subject, assuming a uniform delay of 100ms between neural activity and behavior for all sessions.

After training all methods, we use reconstructed observations from the test trials to evaluate the behavior reconstruction $r^2$. For evaluating the decoding performance from forecasted observations, we used the first 13 time bins (around time till movement onset) to estimate the latent state and embedding, and subsequently use the trained dynamics model to forecast the next 20 time bins. The observations corresponding tp these forecasted trajectories were used to evaluate the prediction $r^2$.

### G.4 HYPERPARAMETERS

**Synthetic Examples**. We used the following architecture for pretraining all methods, with $d_e = 1$ for the Motivating example and Hopf bifurcating system, and $d_e = 2$ for the combined Duffing and Hopf example.

- Inference network
  - $\Omega^i$: $\mathrm{MLP}(d_{y^i}, 64, 8)$
  - $q_\alpha$: $[\mathrm{GRU}(16), \mathrm{Linear}(16, 2 \times d_e)]$
  - $q_\beta$: $[\mathrm{biGRU}(64), \mathrm{Linear}(128, 4)]$
- Generative model
  - $f_\theta$: $\mathrm{MLP}(2, 32, 32, 2)$
  - $h_\vartheta$: $\mathrm{MLP}(d_e, 16, 16, (64 + 33) \times d_r)$
  - $p_{\phi^i}$: $[\mathrm{Linear}(2, d_{y_i})]$
- Training
  - lr: 0.005
  - weight decay: 0.001
  - batch size: 8 from each dataset

**Motor Cortex Experiment**.

- Inference network
  - $\Omega^i$: $\mathrm{MLP}(d_{y^i}, 128, \mathrm{Dropout}(0.6), 64)$
  - $q_\alpha$: $[\mathrm{GRU}(64), \mathrm{Linear}(64, 2 \times d_e)]$
  - $q_\beta$: $[\mathrm{biGRU}(128), \mathrm{Linear}(128, 60)]$
- Generative model
  - $f_\theta$: $\mathrm{MLP}(30, 128, 128, 30)$
  - $h_\vartheta$: $\mathrm{MLP}(d_e, 64, 64, (256 + 158) \times d_r)$
  - $p_{\phi^i}$: $[\mathrm{Linear}(30, d_{y_i})]$
- Training
  - lr: 0.01
  - weight decay: 0.05
  - batch size: 64 trials from 20 datasets

