# OpenReview forum: "Meta-Dynamical State Space Models for Integrative Neural Data Analysis"
_ICLR.cc/2025/Conference — ICLR 2025 Spotlight_

### Official Review · Reviewer_6MTE · 2024-10-23

**Soundness:** 4
**Presentation:** 3
**Contribution:** 3
**Rating:** 8
**Confidence:** 4

**Summary:**

This work proposes a new model class for learning shared sets of latent neural dynamics across heterogeneous data sets. While a number of recent papers have demonstrated empiricially that such population dynamics may be shared across tasks or even species, differences in number of neurons, individual behaviors, etc. present an obstacle to learning a shared model from which inferences might be drawn. The solution in this work involves learning a dataset-specific embedding vector used to parameterize low-rank updates to the weight matrices of state space models. These embedding vectors then parameterize differences across sessions, tasks, and individuals.

The paper is clearly written, and the problem is a timely one for the field. The experiments are thoughtfully chosen and give good insights, though some aspects (data type, model comparisons) are a bit limited. In all, a good paper that's above the bar for acceptance in my view.

**Strengths:**

- Methods for combining insights across datasets are a huge priority in neuroscience.
- The model builds on several well-understood components and proposes a sensible ansatz for how embeddings alter to underlying system dynamics.
- Multiple experiments carefully examine several key variables, including data set size, look-ahead prediction, and number of training trials. These are all clearly presented.

**Weaknesses:**

- The model makes a number of design choices that are unclearly or incompletely motivated. For instance: does the dynamical VAE matter? Should one use the Bowman et al. inference scheme or the SMC-based autoencoder models of [1, 2, 3]? What about the bidirectional RNNs for $\mu_\alpha$ and $\sigma^2_\alpha$? One can't ablate everything, but it would be good to know how much conclusions depend on these choices.
- Related but distinct: The experimental comparisons are done for alternative approaches to learning the embeddings, but there aren't really comparisons with other models. At least one of the metrics tabulated is $k$-step ahead $R^2$, which could surely be compared with LFADS on single-session data or POYO [4] on multi-session data. I recognize that this paper has goals in addition to pure prediction, but since the authors do consider that metric, it would be good to see how the model proposed here compares to other predictive models. Likewise, it would be good to know how to assess (other than qualitatively) what constitutes a "good" learned latent space in the absence of ground truth.

[1] Variational Sequential Monte Carlo
Christian Naesseth, Scott Linderman, Rajesh Ranganath, David Blei Proceedings of the Twenty-First International Conference on Artificial Intelligence and Statistics, PMLR 84:968-977, 2018.

[2] C. J. Maddison, D. Lawson, G. Tucker, N. Heess, M. Norouzi, A. Mnih, A. Doucet, and Y. Whye Teh. Filtering variational objectives. In Advances in Neural Information Processing Systems, 2017.

[3] T. A. Le, M. Igl, T. Jin, T. Rainforth, and F. Wood. Auto- Encoding Sequential Monte Carlo. arXiv:1705.10306, May 2017.

[4] A Unified, Scalable Framework for Neural Population Decoding
Mehdi Azabou, Vinam Arora, Venkataramana Ganesh, Ximeng Mao, Santosh Nachimuthu, Michael Mendelson, Blake Richards, Matthew Perich, Guillaume Lajoie, Eva Dyer

**Questions:**

- There are some experiments to demonstrate that the parameterized low-rank embeddings outperform linear adapters or using embeddings as inputs, and these are convincing, but the intuition supplied for this is pretty terse. Why should one expect interference in these cases? Why does the rank-1 update protect against this?
- Figure 1: It would be helpful if panel B (or some other panel) could include $\Omega$ and the inference networks proposed by the model. For generation, this seems fine, but that's only capturing half the model.
- Is there a ground truth for Figure 2? It's hard to judge what one _should_ be looking for here.

---

> ### Author Response · Authors · 2024-11-18
>
> We thank the reviewer for taking the time to provide thoughtful and constructive feedback on our submission. We are appreciative of the reviewers’ overall positive reception to the work in terms of presentation and experiments, as well as the recognition of the potential impact for the field. We provide clarifications on the points raised by the reviewer below.
>
> > The model makes a number of design choices that are unclearly or incompletely motivated. For instance: does the dynamical VAE matter? Should one use the Bowman et al. inference scheme or the SMC-based autoencoder models of [1, 2, 3]? What about the bidirectional RNNs for $\mu_{\alpha}$ and $\sigma_{\alpha}^2$?
>
> We agree that it is important to expand on the relevance of the chosen inference scheme. In the paper, we followed the standard encoder parameterization introduced in [1] for simplicity, as this is a standard baseline across the dynamical VAE literature. We note that our approach is independent of the parameterization of the encoder and we do not expect the relative performance values reported on the paper to depend on this choice. We will update the manuscript to make this clear.
>
> > The experimental comparisons are done for alternative approaches to learning the embeddings, but there aren't really comparisons with other models.
>
> We agree with the reviewer that in general, comparing across different model classes is important. However, we consider the same model class in our baseline comparisons so as to not conflate the effect of the specific inference scheme with the proposed generative model, which is the novel contribution of our work. It is likely that an improvement in the inference scheme would lead to better performing single-session models. But the results suggest that the corresponding multi-session embedding conditioned generative model would still outperform it in few-shot regimes.
>
> Lastly, although a minor point, we do consider shared dynamics (Session Stitching in LFADS [2], and similar to [3]) and Single session model baselines as well, which do not include embeddings.
>
> > At least one of the metrics tabulated is k-step ahead $R^2$, which could surely be compared with LFADS on single-session data or POYO [4] on multi-session data.
>
> * We could indeed compute this metric by using LFADS on single-session data but we consider the same base model due to the reason mentioned in our previous response. Since LFADS is a special case of a dynamical variational autoencoder, it should be possible to incorporate our generative model into that framework; for instance, by including an additional embedding encoder, and adapting the generative network conditioned on the inferred embedding.
> * Since POYO [4] is trained by minimizing a supervised velocity decoding loss, we would not be able to compute the forecasting $R^2$. For the motor cortex experiments, we used the trained generative model to forecast neural observations starting from movement onset time, and used these forecasted values to decode behavior.
>
> > I recognize that this paper has goals in addition to pure prediction, but since the authors do consider that metric, it would be good to see how the model proposed here compares to other predictive models.
>
> Again, we agree with the reviewer that this would be a good addition. However, we want to highlight that this is the first approach for learning a dynamical systems model from neural recordings that considers non-trivial variations across recordings, to the best of our knowledge.
>
> > Likewise, it would be good to know how to assess (other than qualitatively) what constitutes a "good" learned latent space in the absence of ground truth.
>
> This is indeed an important question for latent variable models of neural data in general. One way of demonstrating that we have learned a good generative model (which includes a good latent space) is being able to forecast activity on novel recordings given a few samples. While good performance on forecasting indirectly implies a “good” learned latent space, this is still very much an open question.
>
> > There are some experiments to demonstrate that the parameterized low-rank embeddings outperform linear adapters or using embeddings as inputs, and these are convincing, but the intuition supplied for this is pretty terse. Why should one expect interference in these cases? Why does the rank-1 update protect against this?
>
> We did not expect these results before running the experiments so we found these observations quite interesting as well. Theoretical results in Galanti & Wolf, 2020 [5] suggest that learning conditional functions using hypernetworks affords improved expressivity compared to a fixed input-based parametrization, and is one possible explanation for our observation. However, this requires further investigation and is beyond the scope of the current work. We will include this in the Discussion section of the updated manuscript.

---

> > ### Comment · Reviewer_6MTE · 2024-11-19
> >
> > I would like to thank the reviewers for their replies and clarifications. I continue to find the work interesting and marginally above the acceptance threshold. However, as nearly all reviewers have pointed out, the lack of model comparisons and consideration of alternatives in the design space remains a weakness. While the authors have argued that these comparisons are inappropriate or out of scope, it is difficult to believe that readers cannot be given _some_ indication of how the predictive performance of the authors' model (which they themselves find important enough to assess) compares with other published work.
> >
> > I plan to maintain my score.

---

> > > ### Author Response · Authors · 2024-11-20
> > >
> > > We thank the reviewer for providing their valuable feedback and helping us improve our submission. We acknowledge the reviewers' existing concerns and are working to incorporate their suggestions in the updated draft of the paper.

---

> > > > ### Author Response · Authors · 2024-11-25
> > > >
> > > > We have uploaded a revised version of the manuscript with additional results and figures taking the reviewers’ feedback into account.
> > > > * Based on the reviewers’ suggestion, we have included **LFADS** with the controller in our behavior reconstruction and forecasting metrics. Although it performs well on reconstruction, it does not have good performance on forecasting, especially in few-shot settings. This further demonstrates the advantage of having a hierarchical model with shared low-dimensional structure for effective few-shot learning.
> > > > * We also verified that our approach is largely agnostic to a particular inference scheme. We trained single session models, as well as our proposed generative model with **Variational SMC** (Details in Appendix D.1, Fig. 12). Reassuringly, the embedding still showed distinct clusters that reflected the underlying dynamical variations across recordings and we were able to use this pre-trained model for learning a generative model given a few trials from new recording sessions.
> > > > * For comparison with an additional model class, we used the **Deep Variational Bayes Filter** (DVBF) as an alternative for inference and learning. Apart from addressing the question of an alternative inference scheme, this also addresses the reviewers’ questions about the specific latent encoder parametrization. While the latent state encoder and dynamics model are independent in the dynamical VAE formulation used in the main text, DVBF explicitly ties them together to improve learning of the dynamical system model. We implement this method using an inference network that encoded observations backward in time as,
> > > >
> > > >     $$q_{\beta}(u_t | \bar{y}_{1:T}, e) = \mathcal{N}(u_t | \mu\_{\beta}(\textrm{concat}[\bar{y}\_{t:T}^i, e]), \sigma^2\_{\beta}(\textrm{concat}[\bar{y}\_{t:T}, e]))$$
> > > >
> > > >     where $q_{\beta}$ was parametrized by a regular RNN to infer the parameters of the Gaussian distribution $u$. The latent
> > > >     state trajectories were sampled as,
> > > >     $$z_t = f_{\theta, e}(z_{t-1}) + Q^{1/2}u_t$$
> > > >
> > > >     We observed similar benefits of our approach on few-shot performance by using this scheme as well (Details in Appendix D.2,
> > > >     Fig. 12).
> > > >
> > > > We thank the reviewer for their suggestions that have helped us further strengthen our paper. We hope that these additional baselines adequately address the reviewers’ concerns about the lack of design choices in our inference network and make a stronger case for our method. If there are any other points we can further clarify or additional experiments we can do to improve the reviewers’ overall assessment of our work, please let us know.

---

> > > > > ### Author Response · Authors · 2024-12-02
> > > > >
> > > > > Thank you again for your valuable feedback. We wanted to kindly ask if the revised version of the manuscript, with the additional comparisons and experiments, has addressed your concerns. With the discussion period coming to an end, we would greatly appreciate it if you could let us know if there are any remaining questions or concerns we can clarify.

---

> > > > > > ### Comment · Reviewer_6MTE · 2024-12-02
> > > > > >
> > > > > > I appreciate the authors’ additional experiments and clarifications. I believe they strengthen the paper. I will revise my score accordingly.

---

> > > > > > > ### Author Response · Authors · 2024-12-02
> > > > > > >
> > > > > > > We thank the reviewer for their positive assessment of our revised manuscript and for raising the score.

---

> ### Author Response · Authors · 2024-11-18
>
> > Figure 1: It would be helpful if panel B (or some other panel) could include $\Omega$ and the inference networks proposed by the model. For generation, this seems fine, but that's only capturing half the model.
>
> We thank the reviewer for the suggestion. We are working on an illustrative figure for the inference framework and will include it in the updated manuscript.
>
> > Is there a ground truth for Figure 2? It's hard to judge what one should be looking for here.
>
> We apologize if the figure is confusing. Since we train the model jointly on multiple datasets (M=2 or M=20), and consequently, multiple underlying dynamical systems, it is difficult to generate a single ground truth figure. We could show separate figures for different datasets, however, it is challenging to discern variations in velocity in vector field plots.
>
> **References**
>
> [1]. Krishnan, R. G., Shalit, U., & Sontag, D. (2015). Deep kalman filters. arXiv preprint arXiv:1511.05121.
>
> [2]. Pandarinath, C., O’Shea, D. J., Collins, J., Jozefowicz, R., Stavisky, S. D., Kao, J. C., ... & Sussillo, D. (2018). Inferring single-trial neural population dynamics using sequential auto-encoders. Nature methods, 15(10), 805-815.
>
> [3]. Herrero-Vidal, P., Rinberg, D., & Savin, C. (2021). Across-animal odor decoding by probabilistic manifold alignment. Advances in Neural Information Processing Systems, 34, 20360-20372.
>
> [4]. Azabou, M., Arora, V., Ganesh, V., Mao, X., Nachimuthu, S., Mendelson, M., ... & Dyer, E. (2024). A unified, scalable framework for neural population decoding. Advances in Neural Information Processing Systems, 36.
>
> [5]. Galanti, T., & Wolf, L. (2020). On the modularity of hypernetworks. Advances in Neural Information Processing Systems, 33, 10409-10419.

---

### Official Review · Reviewer_qSiJ · 2024-11-03

**Soundness:** 4
**Presentation:** 4
**Contribution:** 3
**Rating:** 8
**Confidence:** 3

**Summary:**

The authors present a method for learning the dynamics of neural data across individuals and tasks. Due to the heterogeneity of neural data (e.g. different neurons are recorded across different days, and of course across different subjects) much of the work in inferring dynamics from neural data has focused on single-session models. This limits the ability to train large models with lots of data, and to directly compare dynamics across individuals/sessions. The authors propose a state space model approach that learns a single set of shared latent dynamics, as well as session-level embedding that modifies the shared dynamics. They demonstrate the efficacy of their method on simulated data, and then apply it to real-world neural recordings from monkeys.

**Strengths:**

The paper addresses a long-standing problem in the field of neural data analysis, and one that will only become more pronounced as data collection methods improve. The approach is novel and well-motivated, and provides a tool for not only nicely reconstructing neural dynamics but also providing interpretable model components that will be crucial for scientific understanding. Additionally, the paper is well-written.

The proof of concept presented in figs 2+3 is extremely helpful for introducting the problem, and much appreciated.

The analysis of the synthetic data in sec 5.1 is also well-designed and instructive.

The real neural data is a well-chosen dataset that balances real world heterogeneity and simplicity for exploring model performance and behavior.

**Weaknesses:**

While the authors generally do a great job keeping track of all the notation, I find myself confused about how different trials are handled. It seems that trial averages over, say, certain reach directions are being modeled, rather than individual trials (perhaps I missed this somewhere). But how, then, are the different conditions handled in the notation y_{1:T}^{1:M}? I became confused about this in section 3.2 regarding the read-in network and the dynamical embeddings. Is there a single dynamical embedding per session, or per session-condition? If the former, does this require concatenating data across all conditions? Please clarify. [related, in sec 5.2, are test _trials_ left out, or test _conditions_?]

Fig 5B is difficult to interpret; some more annotations or a clearer caption would be appreciated.

The authors cite large-scale training of models as a motivation for their work, but there are no results that directly demonstrate how adding more data improves performance - the only results here are single session models vs the "all sessions" model. It would be interesting to see some of the metrics computed as a function of the number of sessions used for training the model (and then, for example, testing few-shot performance on held-out sessions/subjects).

**Questions:**

minor comments:
- L390, "Fig" missing before 17?
- would be helpful to show the true dynamics in fig 12

---

> ### Author Response · Authors · 2024-11-18
>
> We thank the reviewer for taking the time to provide thoughtful and invaluable feedback on the manuscript. We are incredibly excited about applying our method to various neuroscientific problems and therefore find the reviewers’ comments highly encouraging. We also really appreciate the reviewers’ positive impressions on the writing and overall presentation of our work. We address the points raised by the reviewer and provide further clarifications below.
>
> > But how, then, are the different conditions handled in the notation y_{1:T}^{1:M}?
>
> We thank the reviewer for bringing our attention to missing details about how trials are handled.
> * We omitted trial labels in the equations in section 3.2 for ease of notation but we are modeling individual trials. We do not include any condition related information in the model (although it can be incorporated as an input), or use the information to get the dynamical embedding. In sec 5.2, the results correspond to performance on left-out trials for each session.
> * Since we focus on differences in dynamics across sessions, we learn a single dynamical embedding per session. We indeed aggregate trials (or batches) that belong to the same session when sampling from $q_{\alpha}$ to get this value.
>
> We apologize for the confusion and are updating section 3.2 to include these important details.
>
> > Fig 5B is difficult to interpret; some more annotations or a clearer caption would be appreciated.
>
> We will update the figure and caption, as well as provide more details on how it was generated in the Appendix in the updated manuscript.
>
> > It would be interesting to see some of the metrics computed as a function of the number of sessions used for training the model (and then, for example, testing few-shot performance on held-out sessions/subjects).
>
> We agree that it would be important to do this analysis. We did include the results of pretraining with 4 sessions, relative to the 44 sessions used in the main text, which supports the relevance of large-scale pre-training on generalization. Specifically, the model pre-trained with 4 sessions generalized quite well to test trials from those sessions (Figure 17B) but it required more trials from novel sessions to achieve the same performance as the “larger-scale” model (Figure 17C vs Figure 6B). We plan to do a more fine-grained comparison of generalization performance as a function of the training data, and data diversity.
>
> > L390, "Fig" missing before 17?
> > would be helpful to show the true dynamics in fig 12
>
> We thank the reviewer for pointing out the typo as well as the suggestion. We will incorporate these in the updated version.

---

> > ### Comment · Reviewer_qSiJ · 2024-11-25
> >
> > I thank the authors for their responses, and am looking forward to some of the clarifications in the manuscript. I maintain my previous rating.

---

> > > ### Author Response · Authors · 2024-11-25
> > >
> > > We have uploaded a revised version of the manuscript after incorporating the reviewers’ feedback. Specifically, we have included additional details in Section 3.2 and expanded on trial handling during embedding inference. We have additionally included a corresponding figure for the inference and read-in networks (Fig. 14).
> > >
> > > We thank the reviewer for their feedback that has helped us improve the clarity and presentation of our work. If there are any other points we can further clarify, do let us know.

---

### Official Review · Reviewer_zU5A · 2024-11-03

**Soundness:** 3
**Presentation:** 4
**Contribution:** 3
**Rating:** 8
**Confidence:** 4

**Summary:**

This paper tackles the problem of analysis across different animals, tasks, and recordings by introducing an unsupervised and dataset-hierarchical model of neural time series data. Their model class seeks to characterize a unified latent space over dynamical structures, which the authors claim has advantages for both inference efficiency and interpretability.  The authors propose different simplifications or ablations to their model class as baselines, showcasing the usefulness of the full interaction between the components. The authors investigate a synthetic example to show recovery performance and provide intuition for the shortcomings of typical approaches. Finally, they analyze recordings in the motor cortex of monkeys performing reaching tasks, highlighting the common dynamics across recordings.

**Strengths:**

- The paper is well-motivated. The introduction frames the paper well by highlighting surrounding literature and results on shared structure across subjects/tasks in task-trained models. Then section 2 provides a compelling example of the limitations of a shared latent dynamics model, with dataset-specificity only in the emission likelihood.
- The paper is generally well-written.
- The synthetic results are strong and form a cohesive story with Section 2
- The motor cortex recording analysis is extensive, analyzing two different tasks, and provides compelling results. In particular, assessing the variability in dynamics across different stimuli and further showing how this variability itself varies between tasks showcases the model's fit. Finally, interpolation in an embedding space (Figure 7) that encompasses both tasks showcases the model's use for interpretability generally.

**Weaknesses:**

While the authors do explore ablations and specific perspectives on their own model class, a comparison to other models is missing. There are two fronts to this comparison:
- **Theoretical and modeling connections**, investigating how their SSM relates to known models. Can you cast known models as special cases of yours? For instance the shared dynamics motifs (Driscoll et al, 2024), how would they best fit within this framework? Alternatively, the embedding analysis of (Cotler et al., 2023), how would it fare on SOTA unsupervised models of neural activity compared to your analysis directly on the data?
- **Empirical comparisons**, implementing the other models mentioned in Section 4 using the way they would typically handle the variability across datasets (such as inputs or indicator variables).

Medium:
- Figures 3 and 4 lack detail on how they were generated, specifically as to which parameters (/embedding values $e^i$) are being used.
- Section 3.1. takes unnecessary turns in presenting the hierarchical model in my opinion. It starts with eq (3) with $\theta^i = \theta + \epsilon_i$, and but then discards this model (due to dimensionality of $\epsilon_i$) in favor of eq. (8) with $e^i \sim p(e)$, only to come back in eq. (11) to a form $\theta^i = \theta + h(e^i)$. I believe a more streamlined exposition could help the reader, going from eq (3) to eq (8) directly in some fashion.

Minor (/typos):
- L131, "we model the likelihood as a linear function of $z_t$": the mean of the likelihood is an affine model (from the $+D$), but the likelihood function/distribution itself is not "linear".
- Eq (5), $\phi^i$ undefined. I gather from L205 that each $\phi^i$ is learned independently  -- I would clarify this aspect.
- L155: I am unsure about the use of the term "inductive bias". $\theta$ and $\epsilon_i$ capture shared and dataset-specific structure *by construction* -- I see the "bias" here as referring to how dynamical structures are similar to a mean structure with $\theta$.
- Eq. (14) and surrounding (and further, L347), $d_r$ and $r$ are used interchangeably

**Questions:**

- How does the read-in network $\Omega$ handle different data dimensionalities?
- A readout matrix $C^i \sim \mathcal{N}(0,0.01)$ is highly contractive. What effect does that have on the model, does it not induce degeneracy?
- What are the color schemes in Figures 3A and 4A? How are these figures generated?
- Why did the linear adapter and Embedding-Input methods get worse as $n_s$ increased in Table 1? Is this from a similar intuition as high M in Section 2?

---

> ### Author Response · Authors · 2024-11-18
>
> We thank the reviewer for taking the time to provide extensive and thoughtful feedback on our submission. We also appreciate the overall positive reception to our work, especially in terms of presentation, and experimental evaluation. We address the reviewers’ concerns below.
>
> > For instance the shared dynamics motifs (Driscoll et al, 2024), how would they best fit within this framework? Alternatively, the embedding analysis of (Cotler et al., 2023), how would it fare on SOTA unsupervised models of neural activity compared to your analysis directly on the data?
>
> This is a great point and we agree that it is important to make the connection between our approach and similar ideas that have emerged in RNNs.
> * There are subtle differences between our work and the dynamical motifs in Driscoll et al., 2024. In the latter, each motif corresponded to a distinct fixed point structure or topology and the context cue was a pre-specified input that could push the dynamics to regions of state space corresponding to task-relevant dynamics. In our case, we want to capture both topological and geometrical differences, and the “context” or embedding is learned from data. In our case, the embedding can induce changes in dynamics even in the same region of state space, i.e., given the same initial condition, dynamics can evolve differently, conditioned on the embedding. However, the broad idea of dynamical structure re-use is similar in both works because after pretraining, rapid learning is facilitated by inferring the embedding of new data, and re-using the corresponding dynamical system model. It would be interesting to reverse engineer the dynamics model learned via our approach to identify potential sub-networks that might more directly relate to the motifs identified in Driscoll et al., 2024 but it is beyond the scope of this paper and we leave such investigations to future work.
> * The embedding analysis in Cotler et al., 2023 is quite similar to our main idea since they also observe that embeddings that are “close” have similar dynamical properties. We believe that the application of both approaches would lead to similar conclusions given sufficient data with high signal-to-noise ratio. However, when there are trial-limited or noisy datasets, learning individual dynamics models for each dataset, as done in Cotler et al., 2023, might be challenging, and leveraging the collective statistical power of all datasets to learn a common dynamical systems model (or meta-model in Cotler et al., 2023) as done in our approach would be preferable.
>
> We hope this offers more insights into the connections between these approaches. We will include a succinct summary of these points in the Discussion section of the updated paper.
>
> > Empirical comparisons, implementing the other models mentioned in Section 4 using the way they would typically handle the variability across datasets (such as inputs or indicator variables).
>
> * Could the reviewer specify which additional methods they would like us to incorporate? The Shared Dynamics baseline incorporates the design choices from Pandarinath et al., 2018 and Herrero-Vidal et al., 2021. The transformer-based approaches only indirectly incorporate temporal information.
>
> * We agree that it is important to compare against different model classes, however, we did not want to confound the effect of the specific inference scheme with our proposed generative model, which is the novel contribution of our work. Moreover, we chose a general sequential variational autoencoder inference framework, but our method can be adapted to work with specific cases as well, for instance, LFADS (Pandarinath et al., 2018).
>
> > Figures 3 and 4 lack detail on how they were generated, specifically as to which parameters (/embedding values $e^i$) are being used.
>
> We will include these details in the updated manuscript.
>
> > Section 3.1. takes unnecessary turns in presenting the hierarchical model in my opinion.
>
> We thank the reviewer for the suggestion. We were trying to tie our model to more classical hierarchical state-space modeling approaches (such as Linderman et al., 2019). We are working on incorporating your comments to make the presentation more concise and streamlined, and will post the updated manuscript soon.
>
> > Minor (/typos)
> L155: I am unsure about the use of the term "inductive bias". \theta and \epsilon^i capture shared and dataset-specific structure by construction -- I see the "bias" here as referring to how dynamical structures are similar to a mean structure with \theta.
>
> We will rephrase and clarify the points mentioned by the reviewer in the updated version.
>
> > How does the read-in network $\Omega$ handle different data dimensionalities?
>
> This network is learned per-dataset (eq. 16, line 223) similar to the likelihood function. As suggested by Reviewer 6MTE, we are working on an illustrative figure for the inference network and will include it in the updated manuscript.

---

> > ### Author Response · Authors · 2024-11-18
> >
> > > A readout matrix $C^i \sim \mathcal{N}(0, 0.01)$ is highly contractive. What effect does that have on the model, does it not induce degeneracy?
> >
> > Thank you for pointing this out; there was a typo in that statement and $C^i \sim \mathcal{N}(0, I/\sqrt{d^z})$. This results in an average prior firing rate of 1.
> >
> > > What are the color schemes in Figures 3A and 4A? How are these figures generated?
> >
> > We apologize for the missing details and inconsistent color schemes. We will add the details in the updated version. These vector fields were generated using the mean dynamical system learned by the model. Briefly, we computed the velocity for uniformly sampled points on a 2-D grid as $f_{\theta + h(e^i)}(z) - z$, with the inferred $e^i$ values. We will expand on this along with the specific values used in the Appendix.
> >
> > >  Why did the linear adapter and Embedding-Input methods get worse as increased in Table 1? Is this from a similar intuition as high M in Section 2?
> >
> > In this experiment, both approaches did not learn the underlying family of dynamical systems as well. As a result, the performance ceiling for these approaches is lower compared to ours and the performance is saturating.

---

> > > ### Author Response · Authors · 2024-11-20
> > >
> > > We thank the reviewer once again for taking the time to provide thoughtful feedback on our submission. We are working on incorporating these suggestions and wanted to clarify what empirical comparisons the reviewer would like us to include in the updated version.

---

> > > > ### Author Response · Authors · 2024-11-25
> > > >
> > > > We have uploaded a revised manuscript with additional experiments as well as modifications on the writing and figures after incorporating the reviewers’ feedback.
> > > >
> > > > We have included a more thorough discussion on the connections between Driscoll et al., 2024 as well as Cotler et al., 2023 and our proposed method in the Related Works section.
> > > >
> > > > We have also added additional baselines which we discuss in our global response to our reviewers. Our previous baseline, **Embedding-Input**, was included to compare against methods that use indicator variables to deal with variability. In addition to this, we have tested the performance of **multi-session CEBRA** (Schneider et al., 2023), a method for inferring latents, which also uses indicator variables to handle variability and included the results in the revised manuscript.
> > > >
> > > > We have added additional details on the figure generation and edited the presentation of the approach in Section 3. Specifically, we have made Section 3.1 more concise based on the reviewers’ suggestions, further expanded on the inference part in Section 3.2, as well as included a corresponding figure for clarity (Fig. 14).
> > > >
> > > > We thank the reviewer for their feedback that has helped us improve the overall presentation of the work. We hope that with these changes and additional experiments, we have been able to adequately address the reviewers’ concerns. If there are any other points we can further clarify to improve the reviewers’ overall assessment of our work, do let us know.

---

> > > > > ### Comment · Reviewer_zU5A · 2024-11-26
> > > > >
> > > > > I would like to thank the authors for largely addressing my questions and updating the manuscript accordingly. I am quite satisfied with the theoretical connections drawn in Sections 4 and Appendix A and the much more thorough incorporation of alternate models (LFADS, CEBRA) and inference methods. My clarifying questions were also addressed (however I still see $d_r$ and $r$ used interchangeably, see L188 -- please fix). I have increased my score.

---

> > > > > > ### Author Response · Authors · 2024-11-28
> > > > > >
> > > > > > We sincerely thank the reviewer for taking the time to carefully review our response and the revised manuscript. We are pleased to hear that the changes and additional empirical evaluations have addressed your questions. We also apologize for not updating the notation for $d_r$ earlier -- the corrected notation is now included in the revised manuscript.
> > > > > >
> > > > > > Thank you again for your valuable feedback and for raising your score.

---

### Official Review · Reviewer_eQP9 · 2024-11-10

**Soundness:** 2
**Presentation:** 1
**Contribution:** 3
**Rating:** 6
**Confidence:** 5

**Summary:**

The proposed meta-learning framework for analyzing neural dynamics across diverse datasets that have shared underlying structure. By encoding dataset-specific variations in a low-dimensional embedding space and conditioning a shared dynamical model on this embedding, it allows for the integration of heterogeneous neural recordings. However, it's essential to consider the model's limitations, such as its dependence on shared dynamics, semi-large-scale pre-training data, and its lack of systematic benchmarking. However, overall, the proposed meta-learning framework presents a promising new direction for neural data analysis.

**Strengths:**

*   **Integration of Multi-Session Neural Recordings:** The proposed framework effectively addresses the challenges of integrating (potentially heterogeneous) neural recordings. It achieves this through a parameterization of latent dynamics using a low-dimensional dynamical embedding to capture variations across datasets. While they are not the first report to do so, it is an important direction/application.
*   **Learning a Shared Dynamical Structure:** The model learns a shared dynamical structure across diverse datasets, which is adapted based on the specific characteristics of each dataset.
*   **Rapid Adaptation to New Data:** The proposed method demonstrates good performance in few-shot learning scenarios, where it can rapidly adapt to new recordings with limited training data. This is particularly valuable in neuroscience research, where collecting large amounts of data can be time-consuming and expensive. The ability to learn from small datasets expands the applicability of the framework and makes it suitable for a wider range of experimental paradigms.
*   **Generalizability:** The framework shows promising results in generalizing to new datasets, as evidenced by its performance on held-out data from both synthetic and motor cortex recordings. By learning a manifold of dynamical embeddings, the model is able to capture the variations in dynamics across different recordings and leverage this knowledge to adapt to new, unseen data.

**Weaknesses:**

**Missing Benchmarks & Issues in Related Works**

- There are some inaccurate statements in the introduction; for example, "CEBRA (Schneider et al., 2023) and CS-VAE (Yi et al., 2022) extract latent representations but do not learn underlying dynamics." Schneider et al. specifically learn the underlying dynamics rather than a priori prescribing them, as other algorithms do (e.g., SIDS, LFADS). CS-VAE assumes the same shared underlying dynamics and uses witching linear dynamical systems (SLDS) to discover them (and the authors should update their citation to: https://elifesciences.org/reviewed-preprints/88602.


- Furthermore, this paper does not provide any benchmarking against models that they themselves acknowledge have the same goal: to create robust encoders across multiple datasets for downstream tasks (e.g., decoding). Therefore, they should report their results in comparison to Ye et al., 2023; Zhang et al., 2023; Caro et al., 2024; Azabou et al., 2024; & Schneider et al., 2023 (which they also fail to cite, but which showed joint training of an encoder rapidly adaptable to unseen data).


 **Dependence on Large-Scale Pretraining Data, and Only Within One Neural Domain:**
- The effectiveness of the proposed framework depends on the availability of large-scale pretraining data. The sources show that models trained on smaller datasets tend to learn specialized solutions that may not generalize well to new recordings. Moreover, they only test on limited motor cortex settings, vs. more complex settings such as vision decoding.

 **Accurate Embedding Inference:**

- The model's generalization ability hinges on accurate inference of the dynamical embedding for a new dataset. If the embedding is not accurately inferred, the model's performance on downstream tasks can be significantly compromised. Showing how corrupted/noisy data, and in other domains (see above) affects the utility of the algorithm will be important for adoption in neuroscience.

**Assumptions of Shared Structure:**
- The current framework assumes that there is a shared structure across the related tasks being analyzed. This assumption might not hold true in all cases, particularly when dealing with recordings from very different brain regions or unrelated behaviors. Future work may need to explore ways to incorporate modularity or other mechanisms to accommodate datasets without expected shared structures. The authors should comment on this.

**Questions:**

See weaknesses.

---

> ### Author Response · Authors · 2024-11-18
>
> We thank the reviewer for their feedback and for acknowledging that the proposed meta-learning approach provides a promising direction for neural data analysis. We address the concerns brought up by the reviewer below.
>
> > **Benchmarks/Related Works**
>
> We apologize if our presentation of these works in the Related Works section was in any way misleading. We believe that this is more of a misunderstanding that we can clarify with better terminology. Broadly, we are interested in building a generative model for neural time series data and there are fundamental differences between our approach and the works mentioned by the reviewer.
>
> > Schneider et al. specifically learn the underlying dynamics rather than a priori prescribing them, as other algorithms do (e.g., SIDS, LFADS).
>
> The reviewer is correct in noting that the authors in CEBRA do not assume a data-generating process since the goal of their work is extracting latent representations for efficient downstream task performance, for instance, decoding behavior. We are instead interested in learning a dynamical systems model [2] of diverse recordings from data. While the self-supervised version of CEBRA conditions the encoding model on time during training, it does not learn a dynamical systems model. We will make this distinction more clear in the revised manuscript.
>
> > CS-VAE assumes the same shared underlying dynamics and uses switching linear dynamical systems (SLDS) to discover them.
>
> In CS-VAE [3], the authors are interested in learning latent variable models from high-dimensional multi-subject behavioral data. We agree that the authors assume shared latent structure and exploit this for training a common RNN behavioral decoder, which broadly aligns with our motivation. However, our focus is on building a generative model for data by learning the underlying dynamical system, which is entirely different. The authors use SLDS post-hoc to find discrete behavioral motifs within trials by treating the learned behavioral latents as observations. Crucially, this analysis is done separately for each session/animal rather than in a unified space. We will update the Related Works section to better highlight these differences.
>
> Lastly, although this is a minor point, both CEBRA and CS-VAE are applied on recordings from different subjects performing the same task. In this case, assuming shared latent dynamical systems also works. We are interested in the case when there are non-trivial changes in the underlying dynamical system, as noted in Section 2 of our manuscript.
>
> > and the authors should update their citation to: https://elifesciences.org/reviewed-preprints/88602.
>
> We thank the reviewer for pointing out the correct citation – we will update it in the revised version.
>
> > Furthermore, this paper does not provide any benchmarking against models that they themselves acknowledge have the same goal: to create robust encoders across multiple datasets for downstream tasks (e.g., decoding). Therefore, they should report their results in comparison to Ye et al., 2023; Zhang et al., 2023; Caro et al., 2024; Azabou et al., 2024.
>
> In the paper, we do acknowledge the similarity with foundation models, specifically in lines 271-272; `our approach shares the same broad goal of pretraining a single model for rapid adaptation on downstream recordings.` However, as noted above, we are *not interested in creating robust encoders across multiple datasets for downstream tasks (e.g. decoding)*. Instead, we want to rapidly learn a dynamical systems model for downstream recordings and consequently, in lines 272-274, we follow up with `these methods leverage transformer-based architectures that encode temporal information indirectly via positional embeddings`. Since transformers do not have recurrence in their architectures, they do not learn a dynamical systems model [2]. We will update the manuscript to better explain this point.
>
> > The effectiveness of the proposed framework depends on the availability of large-scale pretraining data. The sources show that models trained on smaller datasets tend to learn specialized solutions that may not generalize well to new recordings.
>
> While we investigate generalization performance when pretraining with 4 recording sessions, relative to the 44 used in the main experiment, we observe that our method is still able to effectively capture diverse dynamical systems and generalizes well to sessions used for training (Figure 17 A, B), but needs relatively more trials from novel recordings for generalization (Figure 17 C). The improvement in generalization with increase in pretraining data or data diversity is unsurprising and has been observed in other works [4][5][6]. We do not necessarily see this as a weakness of the approach – as data collection techniques are improving, there needs to be concomitant development of methods that can extract meaningful structure from large-scale recordings under varied conditions.

---

> > ### Author Response · Authors · 2024-11-18
> >
> > > Moreover, they only test on limited motor cortex settings, vs. more complex settings such as vision decoding.
> >
> > We certainly agree with the reviewer that it would be a good addition to apply our model on recordings from other brain areas. However, we would like to emphasize that ours is the first approach that can learn diverse dynamical systems in a unified model directly from data. Through extensive experiments on large-scale motor cortex recordings, we have tried to assess the limitations and applicability of our approach. We are excited to apply our approach to recordings from other regions as well, or studying differences in dynamics across areas in the future.
> >
> > > The model's generalization ability hinges on accurate inference of the dynamical embedding for a new dataset.
> >
> > We indeed investigated how well we align to the correct embedding as a function of training samples from novel data (Figure 14) after we obtained the results reported in Table 1. While certainly an important limitation, we have noted this observation in the Limitations and Future Work section (lines 504-506). We do observe that alignment to the correct embedding becomes consistent as we increase samples from novel recordings.
> >
> > > If the embedding is not accurately inferred, the model's performance on downstream tasks can be significantly compromised.
> >
> > In cases when the model converged to a local minima (as shown in Figure 14), it still performs comparable to baselines (see Table 1, $n_s=1$ and $n_s=8$). We have rephrased this statement to clarify this point. We appreciate the reviewers’ suggestion to validate the approach on noise corrupted data and will include this additional analysis in the updated manuscript.
> >
> > > The current framework assumes that there is a shared structure across the related tasks being analyzed…. Future work may need to explore ways to incorporate modularity or other mechanisms to accommodate datasets without expected shared structures. The authors should comment on this.
> >
> > See lines 508-509 in Limitations and Future Work. We agree with the reviewer that this is the natural next step and we are excited to explore this as an extension to our proposed framework. The assumption of shared structure across related tasks is based on previous studies (lines 044-046).
> >
> > **References**
> >
> > [1] Schneider, S., Lee, J. H., & Mathis, M. W. (2023). Learnable latent embeddings for joint behavioural and neural analysis. Nature, 617(7960), 360-368.
> >
> > [2] Durstewitz, D., Koppe, G., & Thurm, M. I. (2023). Reconstructing computational system dynamics from neural data with recurrent neural networks. Nature Reviews Neuroscience, 24(11), 693-710.
> >
> > [3] Yi, D., Musall, S., Churchland, A., Padilla-Coreano, N., & Saxena, S. (2023). Disentangled multi-subject and social behavioral representations through a constrained subspace variational autoencoder (CS-VAE). eLife 12.
> >
> > [4] Ramanujan, V., Nguyen, T., Oh, S., Farhadi, A., & Schmidt, L. (2024). On the connection between pre-training data diversity and fine-tuning robustness. Advances in Neural Information Processing Systems, 36.
> >
> > [5] Azabou, M., Arora, V., Ganesh, V., Mao, X., Nachimuthu, S., Mendelson, M., ... & Dyer, E. (2024). A unified, scalable framework for neural population decoding. Advances in Neural Information Processing Systems, 36.
> >
> > [6] Ye, J., Collinger, J., Wehbe, L., & Gaunt, R. (2024). Neural data transformer 2: multi-context pretraining for neural spiking activity. Advances in Neural Information Processing Systems, 36.

---

> ### Comment · Reviewer_eQP9 · 2024-11-19
>
> Dear authors, thanks for the response. I agree several misunderstandings can be clarified in the text, but I still think there is a large value in benchmarking other methods and clarifying/quantifying what precisely you mean to say you discover the dynamical systems model -- i.e., my understanding is that you are attempting to uncover `pθ()`, in `pθ(zt|zt−1) = N(zt|fθ(zt−1),Q)`, but your metric remains a common downstream task, predicting/forecasting. Let's take Figure 9; the GT and the prediction look wildly different, but your R2 metric is still close to 1; shall I believe you truly learned the *dynamics*, or rather just a good SSM/embedding? That is why I am asking you to compare to other methods that share a common subgoal -- discovering `pθ()` i.e. the latents.
>
> >> "rapidly learn a dynamical systems model for downstream recordings and consequently"
>
> - sorry, what do you mean by rapid here? I can't find any information on time to run your method.
>
> >> We certainly agree with the reviewer that it would be a good addition to apply our model on recordings from other brain areas. However, we would like to emphasize that ours is the first approach that can learn diverse dynamical systems in a unified model directly from data. Through extensive experiments on large-scale motor cortex recordings, we have tried to assess the limitations and applicability of our approach. We are excited to apply our approach to recordings from other regions as well, or studying differences in dynamics across areas in the future.
>
> I just have to disagree here; the motor cortex data is simple and from highly similar tasks across all subjects - at the end of the day, it's motor cortex recordings while animals make simple arm movements.

---

> ### Comment · Reviewer_eQP9 · 2024-11-20
>
> >> learning the dynamical system and discovering the latents involve separate parts of the framework
>
> Yes, exactly. Thanks for being clear here, and this is why I (and other reviewers too) are pushing you to benchmark your solution for finding the latents with other methods/architectures; you could still learn dynamics in this second step. Namely, I’m not convinced your method is *the only way* for latent variable modeling that should be tested in your final estimation of $p_{\theta}( )$.
>
> To note, if your solution worked “on top of” other methods, that would be even more exciting for the field…

---

> > ### Author Response · Authors · 2024-11-20
> >
> > We agree with the reviewer -- we are not proposing a new method for inferring latents, and there are certainly other approaches for performing inference. Could the reviewer clarify what method they would want us to incorporate?

---

> > > ### Author Response · Authors · 2024-11-25
> > >
> > > We have uploaded a revised version of the manuscript where we validate the efficacy of our proposed generative modeling approach with methods for inferring latent trajectories. We performed additional experiments considering the **Deep Variational Baye’s filter** (DVBF) and **Variational Sequential Monte Carlo** (VSMC) approaches. Reassuringly, we observe that both of those methods benefit from the parameterization we propose for learning a generative model in trial-limited regimes (See Appendix D and Figure 12 in the updated manuscript). This demonstrates that our proposed model is agnostic to the overall inference framework.
> > >
> > > We additionally evaluated the performance of multi-session CEBRA by first inferring latent trajectories and then using those inferred trajectories to learn a generative model post-hoc, as suggested by the reviewer. Given that the objective function in multi-session CEBRA identifies invariant features across datasets, we observed that it is not able to effectively handle dynamical variability. As a result, learning a generative model from these trajectories led to poor forecasting performance (See Appendix B and Figure 10 in the updated manuscript). We observed similar performance of this baseline on the experiment with motor cortex recordings. If we were dealing with highly similar or the same behavioral task, invariant features would be able to capture a large amount of variance in the recordings [1]. However, we are additionally interested in capturing the variations in dynamical regimes. Our approach is specifically designed with this goal in mind. In contrast to CEBRA, by jointly performing inference and learning of the generative model, our model can use dynamical information for aligning recordings in the presence of similarities, or learn non-overlapping dynamical features to handle large variations.
> > >
> > > We thank the reviewer for their suggestions which have helped us further strengthen our paper. We hope that with these additional experiments, we have been able to address the reviewers’ concerns and clarify our previous comments. If there are any other points we can further clarify to improve the reviewers’ overall assessment of our work, please let us know.
> > >
> > > [1]. Safaie, M., Chang, J. C., Park, J., Miller, L. E., Dudman, J. T., Perich, M. G., & Gallego, J. A. (2023). Preserved neural dynamics across animals performing similar behaviour. Nature, 623(7988), 765-771.

---

> > > > ### Comment · Reviewer_eQP9 · 2024-11-26
> > > >
> > > > Thanks for your follow up and explanations.

---

> > > > > ### Author Response · Authors · 2024-11-26
> > > > >
> > > > > We hope you will consider adjusting your score if our revisions have adequately addressed your concerns and you now find the paper to meet the standards for acceptance. If there are any remaining issues, we would greatly appreciate your feedback.

---

> > > > > > ### Comment · Reviewer_eQP9 · 2024-12-02
> > > > > >
> > > > > > Thanks for the engagement. I think this paper is marginally acceptable, hence my score of 6. As noted previously, I still would have liked to see more extensive benchmarking. The title suggests a `global solution`, and yet it shows a `narrow application` only on monkey reaching tasks; I feel that should be reflected in the title of the work and the abstract. Thus, while solid, it does not reach the bar of a high-profile paper (oral, spotlight, etc), hence why I am confident in my score of 6.

---

### Author Response · Authors · 2024-11-25

We would like to once again thank all the reviewers for taking the time and effort to provide insightful thoughts and feedback on our submission. A common sentiment from the reviewers was that the manuscript would be strengthened by incorporating additional baseline comparisons. We agree with the reviewers, and motivated by these suggestions, we have now incorporated several more comparisons. We have uploaded a revised version of the manuscript, containing additional numerical experiments along with other modifications to improve the overall clarity and links to related literature in neuroscience. We list all the major updates below,

**Additional method comparisons**

One concern raised by the reviewers was that we do not consider alternative dynamical variational autoencoding models in our baseline comparisons.
* In the revised manuscript, we now include baseline comparisons with the **LFADS** [1] model, one of the most popular approaches for inferring latent dynamics from neural data, on both reconstruction and forecasting in our motor cortex experiments. While LFADS had good performance on reconstruction given enough training data, it does not perform well on forecasting and in settings with limited trials (see Fig. 6A and Fig. 19) – suggesting it has difficulty in learning a predictive model of the dynamics.
* We additionally evaluate latent inference using **multi-session Cebra** [2], by post-hoc learning of a generative model as suggested by Reviewer eQP9. Multi-session CEBRA is designed to extract invariant features across datasets, rather than capture dataset-specific variations. Moreover, since CEBRA does not explicitly model the dynamical system, the latent trajectories inferred from this approach are not smooth and overlap in the state space (Fig. 10A). This was reflected in the dynamical systems model learned from these trajectories, which contained these artefacts and resulted in poor forecasting performance (Fig 10B, Fig 6A).
* We also evaluated our generative model with alternative inference and learning approaches.

    * We evaluate models using the **Variational Sequential Monte Carlo** [3] framework as suggested by reviewer 6MTE. We observed that our approach successfully learned the underlying dynamical system structure which could be leveraged for few-shot generative model learning (Appendix D.1).
    * We also include comparisons to the **Deep Variational Bayes Filter** [4] to evaluate i) an alternative inference scheme and ii) an alternative latent encoder formulation. Since this framework explicitly ties the generative model to the inference network, it performs better than other approaches. However, we observe a similar performance improvement of our proposed generative model on few-shot learning relative to a single session model (Appendix D.2).


**Theoretical/Modeling connections**

We have now included an additional subsection in the Related Works to discuss the connection of our proposed approach to similar ideas that have emerged by studying RNN models of neural activity [5][6][7]. We have also expanded on previously mentioned related works to better communicate the motivation for our work.

**Presentation**

In order to improve the presentation of our method, we have made our explanation of the generative model more concise (Section 3.1). We have also further expanded on the specific inference and learning scheme used in the main text (Section 3.2), and included an additional figure on the inference networks (Figure 14) highlighting the architecture of the read-in networks, $\Omega$, the context encoder, $q_{\alpha}$ and the latent state encoder, $q_{\beta}$.

We would like to thank the reviewers again for their valuable feedback and suggestions. We believe that your suggestions have helped strengthen our manuscript and position it more favorably by including these new baseline comparisons, restructuring for clarity, and linking to other works in the literature.

---

> ### Author Response · Authors · 2024-11-25
>
> **References**
>
> [1] Pandarinath, C., O’Shea, D. J., Collins, J., Jozefowicz, R., Stavisky, S. D., Kao, J. C., ... & Sussillo, D. (2018). Inferring single-trial neural population dynamics using sequential auto-encoders. Nature methods, 15(10), 805-815.
>
> [2] Schneider, S., Lee, J. H., & Mathis, M. W. (2023). Learnable latent embeddings for joint behavioural and neural analysis. Nature, 617(7960), 360-368.
>
> [3] Naesseth, C., Linderman, S., Ranganath, R., & Blei, D. (2018, March). Variational sequential monte carlo. In International conference on artificial intelligence and statistics (pp. 968-977). PMLR.
>
> [4] Karl, M., Soelch, M., Bayer, J., & Van der Smagt, P. (2016). Deep variational bayes filters: Unsupervised learning of state space models from raw data. arXiv preprint arXiv:1605.06432.
>
> [5] Driscoll, L. N., Shenoy, K., & Sussillo, D. (2024). Flexible multitask computation in recurrent networks utilizes shared dynamical motifs. Nature Neuroscience, 27(7), 1349-1363.
>
> [6] Cotler, J., Tai, K. S., Hernández, F., Elias, B., & Sussillo, D. (2023). Analyzing populations of neural networks via dynamical model embedding. arXiv preprint arXiv:2302.14078.
>
> [7] Pellegrino, A., Cayco Gajic, N. A., & Chadwick, A. (2023). Low tensor rank learning of neural dynamics. Advances in Neural Information Processing Systems, 36, 11674-11702.

---

### Meta-Review · Area_Chair_gBkN · 2024-12-19

**Metareview:**

The authors present a meta-learning approach to learning dynamical systems models of neural activity, which can allow for rapid learning of new dynamical systems by training on a large set of other systems. The reviewers were unanimous that this is good work, with many giving it high marks, and so I recommend acceptance.

**Additional Comments On Reviewer Discussion:**

The reviewers all engaged productively with the authors, with many raising their score in response to author rebuttals. Reviewer eQP9 was concerned that the benchmarks were not sufficient, but was convinced to recommend acceptance during the discussion period.

---

### Decision · Program_Chairs · 2025-01-22

Accept (Spotlight)